



# Evaluating High-Resolution Forecasts of Atmospheric CO and CO₂ from a Global Prediction System during KORUS-AQ Field Campaign

Wenfu Tang[1], Avelino F. Arellano[1], Joshua P. DiGangi[2], Yonghoon Choi[2,3], Glenn S. Diskin[2], Anna Agustí-Panareda[4], Mark Parrington[4], Sebastien Massart[4], Benjamin Gaubert[5], Youngjae Lee[6], Danbi Kim[6], Jinsang Jung[7], Jinkyu Hong[8], Je-Woo Hong[8], Yugo Kanaya[9], Mindo Lee[6], Ryan M. Stauffer[10,11], Anne M. Thompson[11], James H. Flynn[12], and Jung-Hun Woo[13]

[1]Dept. of Hydrology and Atmospheric Sciences, University of Arizona, Tucson, AZ, USA

[2]NASA Langley Research Center, Hampton, VA, USA

[3]Science Systems and Applications, Inc., Hampton, VA, USA

[4]European Centre for Medium-Range Weather Forecasts, Reading, UK

[5]Atmospheric Chemistry Observations and Modeling Laboratory, National Center for Atmospheric Research, Boulder, CO, USA

[6]National Institute of Environmental Research, Korea

[7]Korea Research Institute of Standards and Science, Korea

[8]Department of Atmospheric Sciences, Yonsei University, Korea

[9]Japan Agency for Marine-Earth Science and Technology, Japan

[10]Universities Space Research Association, Columbia, MD, USA

[11] Earth Sciences Division, NASA Goddard Space Flight Center, Greenbelt, MD, USA

[12]Department of Earth and Atmospheric Sciences, University of Houston, Houston, TX, USA

[13]Dept. of Advanced Technology Fusion, Konkuk University, Korea

*Correspondence to*: Wenfu Tang (wenfutang@email.arizona.edu)



**Abstract.** Accurate and consistent monitoring of anthropogenic combustion is imperative because of its significant health and environmental impacts, especially at city-to-regional scale. Here, we assess the performance of the Copernicus Atmosphere Monitoring Service (CAMS) global prediction system using measurements from aircraft, ground sites, and ships during the Korea United States Air Quality (KORUS-AQ) field study in May to June 2016. Our evaluation focuses on

CAMS CO and $CO_2$ analyses plus two higher resolution forecasts (16-km and 9-km horizontal resolution), to assess their capability in predicting combustion signatures over East Asia. Our results show a slight overestimation of CAMS $CO_2$ with a mean bias against airborne $CO_2$ measurements of 2.2, 0.7, and 0.3 ppmv for 16-km and 9-km $CO_2$ forecasts, and analyses, respectively. The positive $CO_2$ mean bias in the16-km forecast appears to be consistent across the vertical profile of the measurements. In contrast, we find a moderate underestimation of CAMS CO with an overall bias against airborne CO

measurements of -19.2 (16-km), -16.7 (9-km), and -20.7 ppbv (analysis). This negative CO mean bias is mostly seen below 750 hPa for all three forecast/analysis configurations. Despite these biases, CAMS show a remarkable agreement with observed enhancement ratios of CO with $CO_2$ over Seoul metropolitan and over the West Sea, where East Asian outflows were sampled during the study period. More efficient combustion is observed over Seoul ($\Delta CO/\Delta CO_2$= 9 ppbv/ppmv) compared to the West Sea ($\Delta CO/\Delta CO_2$= 28 ppbv/ppmv). This 'combustion signature contrast' is consistent with previous

studies in these two regions. CAMS captured this difference in enhancement ratios (Seoul: 8-12 ppbv/ppmv, West Sea: ~30 ppbv/ppmv) regardless of forecast/analysis configurations. The correlation of CAMS CO bias with $CO_2$ bias is relatively high over these two regions (Seoul: 0.64-0.90, West Sea: ~0.80) suggesting that the contrast captured by CAMS may be dominated by anthropogenic emission ratios used in CAMS. However, CAMS shows poorer performance in terms of capturing local-to-urban CO and $CO_2$ variability. Along with measurements at ground sites over the Korean peninsula,

CAMS produces too high CO and $CO_2$ concentrations at the surface with steeper vertical gradients (~0.4 ppmv/hPa for $CO_2$ and 3.5 ppbv/hPa for CO) in the morning samples than observed (~0.25 ppmv/hPa for $CO_2$ and 1.7 ppbv/hPa for CO), suggesting weaker boundary layer mixing in the model. Lastly, we find that the combination of CO analyses (i.e., improved initial condition) and use of finer resolution (9-km vs 16-km) generally produce better forecasts.

**1. Introduction**

Anthropogenic combustion significantly impacts air quality, climate, ecosystem, agriculture, and public health at local to global scales (Charlson et al, 1992; Doney et al., 2007; Feely et al., 2004; Heald et al., 2006; Maher et al., 2016). This is especially the case in megacities where human activities are most intense, accompanied by immense energy consumption, mainly in the form of fossil-fuel combustion, which directly leading to enhanced emissions of air pollutants,

greenhouse gases, and waste energy. In particular, cities in the Asian region that are rapidly developing in recent decades are subject to more frequent severe pollution conditions (Yang et al., 2013; Guo et al., 2014; Ohara et al., 2007; Shindell et al., 2008, 2011). It is imperative therefore that we enhance our current capability to monitor, verify, and assess anthropogenic



combustion and its impacts as the number of megacities across the globe is expected to rapidly grow in the following decades (United Nations, 2016). The Copernicus Atmosphere Monitoring Service (CAMS) has a state-of-art global and integrated prediction systems that is currently being implemented to meet this need. The Service is funded by the European Union and it builds upon a legacy of projects such as the Monitoring Atmospheric Composition and Climate (MACC) and

GEMS (Hollingsworth et al, 2008).

For nearly a decade, CAMS has been operationally producing daily global near-real time forecasts and analyses of reactive trace gases, greenhouse gases, and aerosols including global reanalyses and estimation of emissions of these atmospheric constituents (Morcrette et al., 2009; Benedetti et al., 2009; Kaiser et al., 2012; Flemming et al., 2015; Flemming et al., 2017; Massart et al., 2016; Agustí-Panareda et al. 2014, Agustí-Panareda et al. 2017). CAMS global forecasts and analyses are

based on the Integrated Forecasting System (IFS) of the European Centre for Medium-Range Weather Forecasts (ECMWF), which is also used for Numerical Weather Prediction (NWP). CAMS recently developed 2 forecasts at higher resolution, which have potential advantages compared to lower resolution analysis and/or forecast, in terms of local-to-regional air quality (Table 1).

The Korea United States Air Quality (KORUS-AQ) field measurement campaign offers a unique opportunity to

assess the accuracy and consistency of the high resolution forecast and analysis system of CAMS and its skill in simulating atmospheric $CO_2$ from anthropogenic combustion. During May to June 2016, the KORUS-AQ field collected comprehensive measurements of air quality (including $CO_2$ and tracers of fossil-fuel combustion) over the South Korean peninsula and its surrounding waters. KORUS-AQ is an international collaboration between U.S. and South Korea to better understand the factors controlling air quality in the region across urban, rural, and coastal interfaces (Kim and Park, 2014, KORUS-AQ

White Paper). This field campaign follows several NASA-led sub-orbital missions in the past focusing on air quality in the United States (e.g., DISCOVER-AQ, SEAC[4]RS), and pollution outflows from Asia (e.g., TRACE-P, INTEX-B, ARCTAS) and integrating the measurements from these campaigns to satellite retrievals and air quality models (Crawford et al., 2014; Toon et al., 2016; Jacob et al., 2003; Singh et al., 2009; Jacob et al., 2010). Local measurements over the West Sea, often representative of Chinese pollution outflow, and over the Seoul metropolitan area provide a rich dataset that is very useful in

evaluating global prediction and analysis systems like CAMS at city-to-regional scale.

In this study, we evaluate CAMS forecast and analysis of fossil-fuel combustion signatures over the KORUS-AQ spatial and temporal domain. In particular, we use measurements of the main products of combustion (i.e., CO and $CO_2$) from the NASA DC-8 aircraft, along with observations from five ground sites, two research ships, and four satellites to assess the capability of CAMS to monitor anthropogenic combustion. Although CAMS CO and $CO_2$ forecasts and analyses

have been evaluated previously (Agustí-Panareda et al., 2014; Agustí-Panareda et al., 2016; Agustí-Panareda et al., 2017; Claeyman et al., 2010; Massart et al., 2016; Flemming et al., 2009; Flemming et al., 2015; Flemming et al., 2017), this study is unique for the following reasons: (1) This study is a joint evaluation of CO and $CO_2$ species, including their associated enhancement ratios which provide insights on CAMS representation of anthropogenic combustion processes; (2) A focus on megacities provides an important baseline investigation. This is especially the case in East Asia where there is still lack of





detailed information and measurements to constrain emission inventories; (3) KORUS-AQ provides a unique opportunity to evaluate the new high resolution global CAMS forecasts of CO and $CO_2$ at local-to-regional scale. This paper begins with a brief description of CAMS and KORUS-AQ (Section 2), followed by an evaluation of CAMS with airborne measurements (Section 3) and with ground sites, ships, and satellites (Section 4). We provide a summary of our findings in Section 5.

**2. Descriptions of CAMS and KORUS-AQ CO and $CO_2$**

**2.1 CAMS CO and $CO_2$ forecasts and analysis**

The Copernicus Atmosphere Monitoring Service (CAMS) has been providing global forecasts and analysis of atmospheric composition on a daily basis at ECMWF for nearly a decade with applications on air quality and monitoring of long-lived greenhouse gases. CAMS uses the Integrated Forecasting System (IFS) for Numerical Weather Prediction (NWP)

to assimilate a wealth of meteorological observations plus satellite products of atmospheric composition to produce atmospheric analysis of reactive gases (e.g. CO, $O_3$, $NO_2$, $SO_2$), aerosols and long-lived greenhouse gases (e.g. $CO_2$, $CH_4$) on the NWP model grid which are then used as initial conditions to forecast the atmospheric composition with a 5-day lead time. The IFS simulates transport of the chemical species (Flemming et al. 2009, Agusti-Panareda et al. 2017) and includes the on-line integration of modules for atmospheric chemistry (Flemming et al. 2015, 2017) and biogenic $CO_2$ fluxes from

terrestrial vegetation (Boussetta et al., 2013) to model atmospheric composition in conjunction with an assimilation system based on four-dimensional variational (4D-VAR) data assimilation (Rabier et al., 2000). The CAMS global atmospheric analysis and prediction system runs at different resolutions and at a different lag times for the various atmospheric species depending on the use of chemistry in the model and the timeliness of the satellite retrievals used in the analysis. The system providing reactive trace gases and aerosols runs at approximately 40 km horizontal resolution with 60 vertical levels and its

analysis is available less than 1-day behind real time. While higher horizontal and vertical resolution is used for the analysis and forecasts of greenhouse gases. The analysis of $CO_2$ and $CH_4$ is available at around 40 km in the horizontal and 137 vertical levels. Currently the forecasts of $CO_2$ and $CH_4$ have the same resolution as the operational weather forecast at ECMWF (137 levels with 9 km horizontal resolution) but previously their resolution was 16 km (from 2015 to 2016). A CO tracer with simplified chemistry based on a linear CO scheme (Massart et al., 2015) is also available in the high resolution

forecast. However, the $CO_2$ and $CH_4$ analysis is only available 4-days behind real time as the satellite retrievals are not available closer to real time. Because of this, in the 16km resolution forecast $CO_2$, $CH_4$ and linear CO are free running and only the meteorology is initialised with the meteorological operational analysis (see Agusti-Panareda et al. (2014) for further details on free-running forecast configuration). Following a recent improvement in the timeliness of the satellite retrievals, the linear CO is initialised with CO analysis, while $CO_2$ and $CH_4$ are initialised with a 4-day forecast from the $CO_2$ and $CH_4$

40 km analysis in the 9 km forecasts. In order not to lose the small-scale features in the initialization process, a spectral filter is applied to only adjust the large scales in the initial conditions of the forecast (Massart, 2016, personal communication). Table 1 (as well as Fig. S1) provides a summary of the three CAMS configurations and five resulting CAMS products




evaluated in this paper and Fig. 1 depicts the different vertical and horizontal resolutions used in the different CAMS configurations.

For this study, we focus on evaluating the three CO and $CO_2$ forecasts and analysis products listed above, namely; $CO_2$ and CO 16-km forecast (FC16s), analyses (ANs) of $CO_2$ (at 40 km) and CO (at 80 km), and a relatively recent CAMS

9-km $CO_2$ and CO forecast product (FC9s) which are initialized from its respective analysis. The FC9s are different from FC16s in terms of both resolution and initialization as described above (e.g. the FC16s are produced from a free-running simulation of $CO_2$ and CO). The near-real time ANs of CO and $CO_2$ are also different from FC16s and FC9s as these ANs continuously assimilate satellite retrievals of CO total column from Measurements Of Pollution In The Troposphere (MOPITT V5-TIR) and the Infrared Atmospheric Sounding Interferometer (IASI) (Inness et al., 2015) and column averaged

dry-air mole fractions of $CO_2$ ($XCO_2$) from the Greenhouse gases Observing Satellite (GOSAT) (Massart et al., 2016). FC9s CO are initialized from MOPITT and IASI CO analysis at a previous time, which are then downscaled from 80 km to 9 km by a spectral filtering scheme. Due to observational and computing constraints, FC9s of $CO_2$ are initialized and downscaled from a 96-hour forecast of $CO_2$ initialized by GOSAT analysis 4 days earlier.

The IFS contains several components, including an atmospheric general circulation model, a land surface model, an

ocean wave model, an ocean general circulation model, and perturbation models for the data assimilation and forecast (Persson, 2001). Model dynamics and numerical procedures, and physical processes are documented in IFS documentation-Cy43r3 (ECMWF, 2017, https://www.ecmwf.int/search/elibrary/part?title=part&year=2017&secondary_title=IFS). Detailed cloud and precipitation physics of the IFS benefits the calculation of wet deposition (Flemming et al., 2017). As for emissions and surface fluxes, CAMS uses the Global Fire Assimilation System (GFAS) for biomass burning fluxes of $CO_2$

(Kaiser et al., 2012). CAMS uses the anthropogenic $CO_2$ fluxes that are based on the annual mean of the Emission Database for Global Atmospheric Research version 4.2 (EDGARv4.2). As the most recent year available for EDGARv4.2 is 2008, estimated and climatological trends are used to extrapolate to the years after 2008. The land vegetation fluxes for $CO_2$ are calculated online by the carbon module of the land surface model in IFS CTESSEL (Boussetta et al., 2013). A biogenic flux adjustment scheme (BFAS) is employed in CAMS to improve the continental budget of $CO_2$ fluxes (Agustí-Panareda et al.,

2014; Agustí-Panareda et al., 2015; Agustí-Panareda et al., 2016). For CO, CAMS uses anthropogenic and biogenic emissions that are based on the MACC/CityZEN EU projects (MACCity) (Granier et al., 2011), and a climatology of The Model of Emissions of Gases and Aerosols from Nature developed under the MACC (MEGAN-MACC) emission inventories (Sindelarova et al., 2014). GFAS is also used for fire emissions. A linear chemistry scheme is used in FC16s for CO (C-IFS-LINCO) for computationally expediency (Claeyman et al., 2010; Flemming et al., 2012; Massart et al. 2015;

Eskes et al., 2017). Key aspects of the three CAMS configurations evaluated in this study are listed in Table 1.

**2.2 CO and $CO_2$ measurements during KORUS-AQ**

KORUS-AQ is a comprehensive field campaign based on international collaboration between U.S. and South Korea (https://espo.nasa.gov/korus-aq). The goal is to better understand the factors controlling air quality (AQ) in the region across



urban, rural, and coastal interfaces. The field campaign was conducted over South Korean peninsula and surrounding waters from May to June 2016. South Korean peninsula and its surrounding waters is a desirable region to conduct the campaign because: (1) Korea's urban/rural sectors are distinct, which is advantageous for distinguishing anthropogenic and natural emissions; (2) Korea is embedded in a rapidly changing region; (3) the region allows studies of local versus trans-boundary

pollution; and (4) air quality monitoring and ground-based measurements are provided by Korea. AQ measurements (including $CO_2$) from aircrafts, ships, and ground sites were obtained during this period. The campaign was designed to answer three scientific questions, including (1) what are the challenges and opportunities for satellite observations of air quality; (2) what are the factors governing ozone photochemistry and aerosol evolution; (3) how do model performance and needed improvements to better represent atmospheric composition over Korea and its connection to the larger global

atmosphere (Kim and Park, 2014, KORUS-AQ White Paper). Fig. 2 shows the study domain ($30°N - 39°N, 123°E - 133°E$) along with the tracks from DC-8 aircraft flights and research ship deployments. The locations of ground sites are also added in Fig. 2. Satellite retrievals from MOPITT CO and Orbiting Carbon Observatory-2 (OCO-2) $CO_2$ are shown in Fig. 2 to provide spatial context and coverage of remote sensing measurements during the campaign. All the observational data used in this study are summarized in Table 2.

**2.2.1 Airborne CO and $CO_2$ measurements**

We use measurements of $CO_2$ and CO from the DC-8 aircraft. $CO_2$ was measured by Atmospheric Vertical Observations of $CO_2$ in the Earth's Troposphere (AVOCET) using a modified LI-COR model 6252 non-dispersive infrared spectrometer (NDIR). This instrument provides $CO_2$ concentrations with high precision by sensing the difference in light absorption between the continuously flowing sample and reference gases (Vay et al., 2003, 2011;

https://airbornescience.nasa.gov/instrument/AVOCET). $CO_2$ 1 Hz 1σ precision and accuracy are ±0.1 ppm and ±0.25 ppm, respectively. CO was measured by the Differential Absorption CO Measurement (DACOM) instrument via infrared wavelength modulation spectroscopy. The system uses three tunable diode lasers providing 4.7, 4.5, and 3.3 μm radiation for accessing absorption lines of CO, $N_2O$, and $CH_4$. The time response for CO measurements is 1 s; the precision is < 1% or 0.1 ppbv; the accuracy is 2% (Warner et al., 2010; https://airbornescience.nasa.gov/instrument/DACOM). Calibrations for both

instruments were performed during flight at regular intervals using gas standards traceable to the WMO scale ($CO_2$: x2012; CO: x2008) and certified by NOAA ESRL. Details about the two instruments are listed in Table 2. Note that we use the 1 min (60 s) merged DC-8 data in this study. The data is available at NASA Langley Research Center archive (www-air.larc.nasa.gov/missions/korus-aq/).

There were 20 formal DC-8 science flights. Note that for time reference, the 'Date' in this paper refers to the day on

which the flight started in UTC time instead of Korean local time, unless the term 'Local time' is explicitly used. This 'date' in UTC time is one day ahead of Korea local time as all flights typically start at 8am local time. We also divide the flight measurements into five groups based on the land cover below the flight tracks and types of pollution sources with which they can be broadly associated with. These groups are classified as: Seoul metropolitan, Taehwa, West Sea, Seoul-Jeju jetway and





Seoul-Busan jetway (Please refer to Fig. 2 for an illustration of these flight groups). The Seoul metropolitan represents air samples over the large city of Seoul which can have a dominant signature from anthropogenic combustion processes. On the other hand, Taehwa represents air samples over a forest area near Seoul, which can be influenced by both surface carbon fluxes from the local forest as well as anthropogenic emissions from Seoul. Measurements over the West Sea were designed

to capture China pollution outflows. The flight tracks over the West Sea were typically zonal tracks forming a 'wall' between China and Korea (see Fig. 2). These flights are conducted only when a China outflow is expected to be present based on weather and AQ forecasts during the campaign. These measurements enable us to investigate combustion signature from China and differentiate them from Seoul. The Seoul-Jeju jetway and Seoul-Busan jetway groups are two jetway flights on which the DC-8 aircraft frequently obtain measurements. The two jetways are both above the Korean peninsula,

connecting Seoul to Jeju and Busan, respectively. Flights over Seoul-Busan jetway is designed to capture activities in forest, rural, and Busan urban regions. The flights in Seoul-Jeju jetway, on the other hand, samples air over local power plants, transported air from the West Sea, and over nearby croplands. We will discuss our evaluation CAMS for each of these five groups in Section 3.

**2.2.2 Ground-based CO and CO$_2$ measurements**

Observations from the following ground sites are used for comparison with CAMS CO and CO$_2$: Baengnyeong, Fukue, Olympic Park, Taehwa, and Yonsei University (see Fig. 2 for the site locations). The sites in Baengnyeong and Taehwa are managed by the National Institute of Environmental Research (NIER). Baengnyeong site is located in less populated Baengnyeong Island, Incheon which is northwest of Seoul. The Fukue site belongs to the Japan Agency for Marine-Earth Science and Technology (JAMSTEC) and is located on remote island of Fukue, Japan. The Olympic Park and

Yonsei University sites belong to Korea Research Institute of Standards and Science and Yonsei University, respectively. Both sites are located within the Seoul Metropolitan area. These five ground sites cover different environments, which allows us to differentiate between urban (Olympic Park and Yonsei University) and remote (Baengnyeong and Fukue) air quality conditions during the campaign. The sites in Baengnyeong, Fukue, and Olympic Park provide measurements of CO (in ppbv), while the site in Yonsei University provides measurements of CO$_2$ (in ppmv). Only the site in Taehwa provides

measurements of both CO (in ppbv) and CO$_2$ (in mg/m$^3$) (Kim et al., 2013). Locations of the five sites, and corresponding instruments and data intervals are provided in the Table 2. Note that we use data from these sites taken during the KORUS-AQ campaign period to provide the ground context of our evaluation.

**2.2.3 Ship observations**

We use ship measurements of CO from Jangmok and Onnuri. Both of them are research vessels owned by Korea

Institute of Ocean Science and Technology. The ship deployments are part of the Korea-United States Ocean Color (KORUS-OC) field study coinciding with KORUS-AQ. KORUS-OC was led by NASA and the Korean Institute of Ocean Science and Technology, focusing on the ocean color, biology and biogeochemistry as well as atmospheric composition in



coastal waters adjacent to Korea (https://www.asp.ucar.edu/sites/default/files/4_Emmons_07_27_2016.pdf). The two ships sailed along the Korean coast from May 20th to June 5th. Tracks of the two ships are shown in Fig. 2 by dark grey (Jangmok) and light grey (Onnuri). CO measurements in Jangmok and Onnuri were taken from the Thermo 48i-TLE CO analyzer and Thermo 48C CO analyser, respectively (http://www.kiost.ac.kr/kor.do), and are provided every minute.

**2.2.4 Satellite-derived CO and $CO_2$ retrievals**

We use four sets of satellite-derived measurements for comparison with CAMS CO and $CO_2$. We use retrievals of $CO_2$ column-averaged dry air mole fraction ($XCO_2$) from NASA OCO-2, version 7, Level 2 (L2) full product with recommended quality control (Crisp et al. 2004; Boesch et al., 2011; Wunch et al. 2011a, b; https://oco.jpl.nasa.gov/). and from Japan Aerospace Exploration Agency (JAXA) GOSAT, Level 2 (L2), version 2 (Yokota et al., 2004, 2009; Morino et

al., 2011; Crisp et al. 2012; http://global.jaxa.jp/projects/sat/gosat/). Short-wavelength Infrared observations measured by the Thermal And Near-infrared Sensor for carbon Observation (TANSO) onboard the GOSAT satellite are used to retrieve $XCO_2$. OCO-2 also has three specific Near Infrared (NIR) wavelength bands to retrieve $XCO_2$ (https://oco.jpl.nasa.gov/). For CO, we use the NASA Terra MOPITT version 6, Level 2, multispectral (Thermal Infrared/Near Infrared; TIR/NIR) total column retrievals (MOP02J, L2, V6) with recommended quality control. Compared to thermal infrared only retrievals (TIR),

these retrievals have an enhanced sensitivity to the lower tropospheric CO (Deeter et al., 2014; https://www2.acom.ucar.edu/mopitt). In addition, we also use total column mole fractions of CO from IASI, Level 2 data with recommended quality control (George et al., 2009; Clerbaux et al., 2009). IASI is on board MetOp-A and B satellites and uses Fast Optimal Retrievals on Layers for IASI (FORLI) to retrieve CO distributions from the thermal infrared (TIR) spectra. We applied the associated averaging kernels from MOPITT and IASI to CAMS CO before comparison as these

retrievals exhibit large sensitivities in the free troposphere. We also note that both IASI and MOPITT have significantly more observations than OCO-2 and GOSAT. As summarized in Table 2, resolutions of OCO-2, GOSAT, MOPITT, and IASI are 2.25×1.29 km, 10.5×10.5 km, 22×22 km, and 12×12 km, respectively. The revisit times for the four satellites are also different. OCO-2 overpasses at 1:18 - 1:33 pm, GOSAT overpasses at around 1 pm. Revisit time is 10:30 am for MOPITT, and 9:30 am for IASI. Uncertainties have also been reported for these satellite products. OCO-2 $XCO_2$ has

uncertainties of 1-2 ppm (Boesch et al., 2011) while GOSAT $XCO_2$ has retrieval errors of 2 ppm (Griffith et al. 2011; Crisp et al. 2012). Deeter et al. (2014) reported 0.09e18 molecules/$cm^2$ for total column retrieval for MOPITT. Wachter et al. (2012) reported uncertainties to be <13% for IASI FORLI.

**3. Comparison with airborne measurements**

Here, we evaluate CAMS forecasts and analysis of CO and $CO_2$ with NASA DC-8 aircraft observations. We
interpolate the 4-D fields of CAMS CO and $CO_2$ model output to collocate with flight measurements in both space and time. The equivalent model data for all flights and for the three configurations (FC16s, FC9s, ANs) are made available in the same file format as the 1-min merged DC-8 dataset to facilitate model to observation comparison. We also estimate enhancement



ratios of CO and $CO_2$ from both airborne and model data and analyse its spatial and temporal variations across different flights. We present in the following subsections the summary statistics of our comparison of CAMS data with DC-8.

### 3.1 Performance across all flights

Across all flight data, CAMS overestimates $CO_2$, with mean biases of 2.2, 0.7, and 0.3 ppmv for FC16s, FC9s, and
ANs, respectively. As found by Agusti-Panareda et al (2016), the overall overestimation of $CO_2$ associated with the biogenic bias correction. In contrast, CAMS underestimates CO with mean biases for FC16s, FC9s, and ANs against DC-8 of -19.2, -16.7, and -20.7 ppbv, respectively. The mean bias is calculated as the average across all data of CAMS minus DC-8. We also find that the overall pairwise correlation between DC-8 and CAMS is moderately high ($CO_2$: 0.52 to 0.57, CO: 0.65–0.73) while the root-mean-square-errors (RMSEs) in CAMS relative to DC-8 are about 7 ppmv for $CO_2$ and 80 ppbv for CO.
These statistics can be summarized using a Taylor diagram as shown in Fig. S2 and Fig. S3 of the supplementary material. We also calculated the associated Taylor scores to summarize the skill of CAMS in capturing the observed $CO_2$ or CO variations (please refer to Equation S1 in the supplement). We find that CAMS have relatively good skill regardless of configuration: for $CO_2$, the skill scores are 0.82 (FC16s), 0.82 (FC9s), and 0.75 (ANs); while for CO, the skill scores are 0.85 (FC16s), 0.86 (FC9s), and 0.83 (ANs). However, it is important to note that these statistics can vary from flight to flight
and the skill for $CO_2$ is not necessarily related to those for CO. For instance, for the May 10th flight, where a southern peninsula outflow was expected, CAMS ANs show higher skill than those from FC9s in terms of both $CO_2$ and CO, while the scores of FC16s are higher than those of FC9s in terms of CO (Fig. S4). Yet, for the May 3rd flight, where a weak Chinese influence was expected, the scores of FC16s and FC9s are higher for $CO_2$ than CO, while we find the opposite for the June 2nd flight, where DC-8 sampled local influences. Lastly, we note that the skill of CAMS during the June 4th flight is
not high for either species. This flight was designed to measure local point sources with large variations at much finer scales.

### 3.2 Performance across individual flights

We present in Fig. 3 the summary statistics of CAMS against DC-8 measurements for all 20 individual flights. This is shown in the second to fourth rows of Fig. 3 as boxplots of the bias for FC16s, ANs and FC9s, respectively. We also show the boxplot of DC-8 $CO_2$ (first row left column) and CO (first row right column) for each flight as points of comparison. The
overall mean, median, interquartile range (IQR), and standard deviation (sigma) of DC-8 $CO_2$ mixing ratios (in ppmv) are 410.37, 408.25, 5.97, and 7.73 respectively. The overall mixing ratio, which varies within 1 to 2 percent, are slightly higher than the month median observed in Mauna Loa (NOAA https://www.esrl.noaa.gov/gmd/ccgg) for May 2016 (408±1 ppmv). For DC-8 CO mixing ratios (in ppbv), the corresponding statistics (mean: 204.59, median:183.90, IQR:127.97, sigma: 101.74) show enhanced CO (and larger variance) than the background value observed in Mauna Loa (100±24 ppbv). In
general, CAMS overestimates $CO_2$ and underestimates CO for most flights. Differences also exist among the 20 flights in terms of both DC-8 measured mixing ratios and model biases. For flights with higher observed variances, CAMS biases and the corresponding variance of the biases tend to be also larger. This is related to variations in weather conditions during the



campaign along with variations in sampling goals of the science flights. For example, the flights in May 3rd, May 17th, May 24th, May 29th, and May 30th were specifically designed to capture Chinese pollution outflow. In these days, the variances in CAMS biases for CO (but not $CO_2$) are larger than the average. The colored shades in Fig. 3 indicate flights for 'special conditions'. The grey and yellow shades indicate two special cases that we study in detail in later sections. In particular, DC-8 flew a 'wall' over the West Sea on May 24th to investigate the transport of Chinese pollution. On June 4th, DC-8 flew near Seoul to measure pollution from local point sources (e.g., power plants). The other shades indicate that the flights were conducted during a frontal passage (purple) and that the flights may possibly be affected by fires in Siberia (orange). These flights were not further analyzed in this study since for example the May 26th flight (with frontal passage influence) and the May 17th and May 19th flights (with possible fire influence) do not clearly stand out from the other flights (see Fig. 3).

### 3.3 Performance across flight groups

Here, we evaluate CAMS per flight group as described in section 2.2.1. We show in Fig. 4 the probability density functions (pdfs) of CO and $CO_2$ for DC-8 and CAMS per flight group. The pdf of CAMS $CO_2$ (which exhibits a longer tale to higher values) show a general offset to higher values relative to DC-8 (except for West Sea). There is a systematic overestimation of CAMS $CO_2$ against DC-8. Accordingly, the 'apparent local background' of $CO_2$ (lower tales of the pdf) is relatively high in CAMS than DC-8. In contrast, CO is underestimated in CAMS across all of the five groups. The pdfs of CO in CAMS show a bi-modal distribution (except in Taehwa and West Sea) indicative of two dominant AQ conditions sampled by DC-8 over this region. The shapes of the CO pdfs in CAMS largely differs from DC-8 (except in Taehwa). We see a higher frequency of occurrence of the two to three modes in West Sea in CAMS that is not apparent in DC-8 while the opposite is the case in Seoul-Busan. This suggests that the underestimation of CO in CAMS may not be systematic or may be caused by biases in CO background values. The pdf over the West Sea also show that CAMS underestimates (or even misses) the more elevated CO observed in DC-8.

We further investigate the differences between CAMS and DC-8 by looking at the bias in the mean profiles. We show in Fig. 5 the mean profiles for all data and each individual group. We find that the overall bias in CAMS $CO_2$ is systematic and close to uniform across all layers (FC16s: ~2.2 ppmv, FC9s: ~1 ppmv, and ANs: ~0.8 ppmv). This overestimation is true for all flight groups except over West Sea. On the other hand, for CO, the overall bias in CAMS is mostly evident in the lower troposphere (about -20 to 25 ppbv below 700 hPa). This underestimation is especially the case over the West Sea and is consistent with the pdfs in Fig. 4.

### 3.3.1 Seoul metropolitan and Taehwa

The airborne measurements over the Seoul metropolitan area were mostly during frequent aborted landing maneuvers (i.e. missed approaches) over Seoul air base. More than 90% of the measurements in this group are taken below 850 hPa. Fig. 4 shows that the performance of FC16s, FC9s, and ANs are alike over Seoul for both CO and $CO_2$, in contrast to the other four flight groups. Given that the measurements over Seoul are dominated by boundary layer and anthropogenic





emissions in Seoul, the model performance over Seoul are most likely to be driven by local emissions. We show in Fig. 6 the mean vertical profiles over Seoul below 800hPa. For $CO_2$, FC9s profiles agree well with the observations. This is not the case for CO, where FC16s, FC9s, and ANs do not agree well with DC-8, but with the bias in ANs being relatively smaller. However, the near surface temporal variations (changes in the profile from morning to afternoon) observed by DC-8 are

captured by FC16s, FC9s, and ANs. It is worth noting that over Seoul, there is an abrupt change in the profile at around 925 hPa for both CO and $CO_2$ of the morning samples. Accordingly, CO is overestimated below 925 hPa and underestimated above 925 hPa. This vertical gradient (i.e., change in mixing ratios divided by change in pressure) in the morning samples of DC-8 $CO_2$ and CO are about 0.25 ppmv/hPa and 1.7 ppbv/hPa, respectively. In contrast, the gradients of $CO_2$ in CAMS are 0.50 ppmv/hPa for FC16s, 0.34 ppmv/hPa for FC9s, and 0.45 ppmv/hPa for ANs while the gradients of CO in CAMS are 4.2

ppbv/hPa for FC16s, 3.4 ppbv/hPa for FC9s, and 3.3 for ANs. It is evident that these gradients (CO and $CO_2$) regardless of CAMS configuration are significantly steeper than observed. While in part this may be attributed to overestimation of emissions during rush hours (and night-time) in Seoul along with model representativeness errors in the boundary layer, we attribute this steep gradient to a possible weaker boundary layer mixing in CAMS since there is an important contrast between near surface CO (overestimation) and CO aloft (underestimation) which cannot be explained by emissions alone.

This is not very apparent in $CO_2$ since there is an overestimation of background $CO_2$ superimposed on this difference.

In Taehwa, the differences between morning and afternoon samples are not as large compared to Seoul metropolitan. The $CO_2$ profiles from ANs and FC9s are apparently closer to DC-8 than from FC16s. However, this difference is not obvious for the CO profiles. Note that in the afternoon (2-4pm), measured $CO_2$ mixing ratio near surface (at 975 hPa) becomes lower than the layer above, indicating a possible drawdown of $CO_2$ by underlying vegetation in Taehwa.

This change is captured by CAMS, especially in FC9s.

### 3.3.2 West Sea

As previously mentioned, the flights over the West Sea are focused on capturing pollution outflow from China. Both CO and $CO_2$ in this flight group are underestimated by CAMS below 900 hPa (Fig. 5). It is the only group in which near surface $CO_2$ is underestimated by all the three CAMS configuration. In addition, the underestimation of CAMS CO over

the West Sea is more significant than that over the other groups. We list two possible reasons for this unique model performance over the West Sea considering that the Chinese outflows constitute the dominant influence of CO and $CO_2$ samples in this group. First, the transport of surface pollution from China to the West Sea is not well represented in CAMS. Second, emissions in China may not be as well quantified than in Korea. During the May 24[th] flight, a strong outflow from China was expected, so DC-8 aircraft flew an extended sampling "wall" over the West Sea to sample transport from China.

We show in Fig. 7 some of the details of this flight. In particular, we show the vertical cross sections of meridional (panel a) and zonal (panel b) fluxes of CO and $CO_2$ in CAMS FC9s. These fluxes are calculated as the product of meridional (from west to east) or zonal (from south to north) wind speed with simulated species density (i.e. in terms of units, $\frac{m}{s} \times \frac{kg}{m^3} = \frac{kg}{m^2 \cdot s}$).





The China outflow moving towards West Sea and Seoul is well demonstrated in the fluxes of CO in panel (a) and (b) especially in the region marked by the black rectangles. This outflow is not apparent in the fluxes of $CO_2$. This is because the variations in $CO_2$ density are very low relative to $CO_2$ background in contrast to CO variations. Hence, the wind speeds dominate the transport flux variations in $CO_2$. We also show in Fig. 7 panel (c) the measurements of DC-8 aircraft and the

bias of FC9s over the West Sea on that day. As can be seen in Fig. 7, CAMS $CO_2$ and CO are largely underestimated ($CO_2$: 2-4 ppmv, CO: 86-88 ppbv) for this flight. This underestimation in both species is consistent with Fig. 5. Note that the underestimation of $CO_2$ over the West Sea is not consistent with other flights and the overall results. This underestimation could be associated with an underestimation of anthropogenic emissions in China, and/or transport from China to the West Sea. This is discussed in Section 3.4 in more details. In summary, the transport pattern of China outflow (CO and $CO_2$) to the

West Sea is captured but the abundances of both CO and $CO_2$ are underestimated by CAMS especially near the surface.

### 3.3.3 Seoul-Jeju and Seoul-Busan jetways

Measurements in the Seoul-Jeju and Seoul-Busan jetways are both above the South Korean peninsula, connecting Seoul to Jeju and Busan, respectively. While both flight groups share some common features, they are treated here as two distinct groups for the following reasons: (1) Seoul-Jeju jetway is close to the west coast of South Korea, whereas Seoul-

Busan jetway sampled air southeast of Seoul and more inland; (2) There are more croplands, urban, and build-up areas along Seoul-Jeju jetway while there are more forested areas along Seoul-Busan jetway; (3) There are some important point sources along Seoul-Jeju jetway such as power plants and the Daesan chemical facility. In fact, the June 4[th] flight was designed to survey point sources west of Seoul and focused more to the Seoul-Jeju jetway. Details of the June 4[th] flight are summarized in Fig. 8. In contrast to the overall statistics across all flight groups, FC16s, FC9s, and ANs for this flight clearly

overestimate CO near point sources. We also note that measurements for this flight are mostly taken below 900 hPa. As such, the spatial variations are larger near point sources than in other conditions. Nevertheless, these variations are not well captured by CAMS, especially by ANs. This may be due to its coarser grid representation (i.e., 40 km for $CO_2$ and 80 km for CO). In addition, we find a difference in terms of mean bias in $CO_2$ between CAMS FC9s and FC16s. This difference is not apparent in CO. This implies there might be large spatiotemporal errors existing in CO emission inventories in the region,

since higher emission resolution does not result in an improvement. In this case, increasing the spatiotemporal resolution might even weaken the simulation results, whereas lower resolution usually agrees better with observations as it "diffuses" the error of the emissions.

### 3.4 Enhancement ratios of CO to $CO_2$

We also evaluate the three CAMS configuration against DC-8 in terms of enhancement ratios of CO to $CO_2$ for all

flights and for each flight group. We conduct a reduced major axis (RMA) regression to estimate the sensitivity of CO to $CO_2$ ($\Delta CO/\Delta CO_2$) with the 1 minute merges. We use RMA instead of ordinary least squares (OLS) regression as the two variables (CO and $CO_2$) are both subject to error (Smith, 2009). The slope estimate in the RMA corresponds to enhancement




ratio of CO and $CO_2$. This ratio can reflect the emission ratios of a particular area especially when using near field data (Parrish et al. 2002). Such analysis has been used in previous studies for surface CO and $NO_X$ (Parrish et al. 2002), flask samples of CO and $CO_2$ in East Asia (Turnbull et al., 2011), airborne measurements of CO and $CO_2$ during TRACE-P (Suntharalingam et al. 2004), surface CO and $CO_2$ in rural Beijing (Wang et al. 2010) and more recently with satellite

retrievals of CO (MOPITT) and $CO_2$ (GOSAT) (Silva et al., 2013). We present our estimates of $\Delta CO/\Delta CO_2$ (with units of ppbv/ppmv) from DC-8 and CAMS FC16s, FC9s and ANs in Table 3. Overall, the observed $\Delta CO/\Delta CO_2$ during the KORUS-AQ campaign is ~13 ppbv/ppmv (or ~1.3%). This is a relatively low value compared to reported ratios in more polluted megacities such as Beijing. The lowest $\Delta CO/\Delta CO_2$ among the five flight groups is observed over Seoul (~9 ppbv/ppmv). The observed $\Delta CO/\Delta CO_2$ for other groups within Korea ranges from ~10 ppbv/ppmv (Seoul-Jeju) to ~16 ppbv/ppmv (Seoul-

Busan and Taehwa). Taehwa is close to and sometimes downwind of Seoul, but has higher observed $\Delta CO/\Delta CO_2$ than Seoul. We attribute this difference to biogenic CO sources and biospheric influence on $CO_2$ over Taehwa. The highest $\Delta CO/\Delta CO_2$ (~28 ppbv/ppmv) is observed over the West Sea. This ratio is a sharp contrast to Seoul and other flight groups over Korea. This indicates that the bulk combustion efficiency over Seoul is higher in Seoul than in the China pollution outflows over the West Sea. The ratio over the West Sea is very consistent with $\Delta CO/\Delta CO_2$ observed over China (upwind of West Sea) during

KORUS-AQ by ARIAs (20-100 ppbv/ppmv (REF). Such 'combustion signature contrast' is consistent with previous studies in the region. During TRACE-P in 2001, the observed ratio over Japan is ~12-17 ppbv/ppmv and ~50-100 ppbv/ppmv over northern China (Suntharalingam et al. 2004). Over Shangdianzi, China and Tae-Ahn Perninsula (TAP), Korea, Turnbull et al. (2011) reported $CO:CO_2$ff ratios (which are derived from measurements of CO and $\Delta^{14}CO_2$ in flask samples taken during winter 2009/2010), of ~47 and ~44 ppbv/ppmv, respectively. They also reported that the South Korea samples from TAP

have $CO:CO_2$ff of ~13 ppbv/ppmv. Wang et al. (2010) reported a change in observed $\Delta CO/\Delta CO_2$ near Beijing from 34-42 ppbv/ppmv in 2005-2007 to 22 ppbv/ppmv in 2008. Finally, $\Delta CO/\Delta CO_2$ derived from satellite retrievals in 2010 indicate a similar contrast between Beijing/Tianjin (~25-50 ppbv/ppmv) and Seoul (~7-9 ppbv/ppmv). Despite the differences in the data sources (satellites, airborne measurements, flask samples) and time period, these $\Delta CO/\Delta CO_2$ values are consistent and all point to a 'combustion signature contrast' between Korea and China. We expect that this contrast may be decreasing over

time as Chinese combustion activities become more efficient.

These observed ratios are remarkably consistent with $\Delta CO/\Delta CO_2$ from CAMS (see Table 3). The three CAMS configurations have $\Delta CO/\Delta CO_2$ over Seoul metropolitan of ~8 to 12 ppbv/ppmv and over West Sea of ~31-32 ppbv/ppmv. Our rough estimates of CO to $CO_2$ emission ratios in CAMS over Seoul and China during KORUS-AQ also show marked similarity with CAMS enhancement ratios. The CO to $CO_2$ emission ratios over China is about 28 (1000 mole/mole) and

about 10 (1000 mole/mole) over Korea. Our results suggest that CAMS emission ratios reflect this contrast and that the modeled $\Delta CO/\Delta CO_2$ is indicative of emissions of Seoul and China. To further understand the skill of CAMS in capturing this contrast, we compare the observed correlation between CO and $CO_2$ and the correlation from CAMS FC16s, FC9s, and ANs. This corr($CO_2$,CO) is presented in the second row of Table 3. Over Seoul, the observed corr($CO_2$,CO) is moderately high



(~0.8), which is likely driven by common CO and $CO_2$ sources (mostly local anthropogenic emissions from Seoul). This correlation is well captured by ANs and FC9s but not FC16s. We attribute this difference to a better initialization in ANs and FC9s due to assimilation. The observed corr($CO_2$,CO) over the West Sea even higher (0.89), indicating that CO and $CO_2$ comes from common sources in China. However, this corr($CO_2$,CO) is not captured by any of the three configurations (0.25-

0.42). A few factors may contribute to this low corr($CO_2$,CO) over the West Sea. First, the flight on May 12[th] is a noteworthy source of low corr($CO_2$,CO) in CAMS. We have shown in Fig. 3 that the major goal of this flight is to study AQ conditions during a frontal passage instead of sampling China outflows. Even though part of the track during May 12[th] is located in the West Sea, the AQ features of that day are evidently different from China outflow events. After excluding measurements during May 12[th], the corr($CO_2$,CO) in CAMS (FC16s-0.51, FC9s-0.43, and ANs-0.29) are now higher albeit still lower than

observed (0.9). Uncertainties in model transport can be a likely cause as the corr($CO_2$,CO) can be subject to transport errors even though $\Delta CO/\Delta CO_2$ may not necessarily be affected. Performance of CAMS over Baengnyeong site (discussed in Section 4.1) also implies possible issues with transport of China pollution towards the West Sea. Furthermore, the difference in temporal representation of China emissions in CAMS may contribute to this mismatch in timing and hence resulting to low correlation. As mentioned in Section 2, CAMS uses prescribed monthly emission for CO while the diurnal cycle of $CO_2$

fluxes is calculated online in CAMS. Lastly, the corr($CO_2$,CO) in FC16s and FC9s are closer to observed corr($CO_2$,CO) than in ANs suggesting that resolution may also play a role. For the other three flight groups, the observed corr($CO_2$,CO) are not as high as those over Seoul and the West Sea. This implies that $CO_2$ and CO observed over these three flight groups may not come from common sources and/or have been mixed with the environment. CAMS corr($CO_2$,CO) do not always agree with observed corr($CO_2$,CO). Overall, corr($CO_2$,CO) from FC16s is higher than observed while corr($CO_2$,CO) from FC9s and ANs

agree well with observed corr($CO_2$,CO). Again, this may be related to the fact that FC16s comes from a free running simulation (i.e., not initialized with analyses).

   Finally, we present the correlation between the biases of CAMS for the two species (corr($Bias_{CO}$,$Bias_{CO_2}$)) (please see the third row of Table 3). This correlation provides another piece of information on whether the performance of CAMS in $CO_2$ and CO are related. We find that corr($Bias_{CO}$,$Bias_{CO_2}$) are high over Seoul and the West Sea, indicating that the

performance of CAMS in CO and $CO_2$ are related for the two groups. Over the West Sea, FC16s, FC9s, and ANs perform similarly. However, the corr($Bias_{CO}$,$Bias_{CO_2}$) are lower in the other three groups relative to Seoul and the West Sea. In addition, our results show that ANs and FC9s usually have lower corr($Bias_{CO}$,$Bias_{CO_2}$)) than FC16s, especially over Seoul. This implies that FC16s performance in $CO_2$ and CO are more strongly related than in FC9s and ANs performance, which could be associated again with the fact that FC16s comes from a free running simulation while FC9s and ANs are both

initialized from analyses. The assimilation of CO and $CO_2$ satellite retrievals may reduce the interdependence of CAMS $CO_2$ and CO performance.





## 4 Comparison with other measurements

In this section, we evaluate CAMS FC16s and FC9s, and ANs against CO and/or $CO_2$ measurements from five ground sites, two ships, and four satellites. Unlike DC-8, data on $CO_2$ or CO in these cases may not be jointly available. In particular, each ground site (except Taehwa) only measures one of the two species. The ships also provide measurements for

CO only while the four sets of satellite retrievals of $CO_2$ and CO are from four different instruments on board four different satellites. Therefore, in this section, $CO_2$ and CO are evaluated separately, and relationships between $CO_2$ and CO inferred from some of these sites are only indicative of a larger pattern that we see in DC-8.

### 4.1 Comparison with ground observations

Here, we focus our evaluation on CAMS performance in capturing surface conditions and diurnal cycle of $CO_2$

and/or CO. Data from the following five ground sites are used in this study: Baengnyeong, Fukue, Olympic Park, Taehwa, and Yonsei University (Fig. 2 and Table 2). It can be seen in Fig. 9 that CO from Olympic park and $CO_2$ from Yonsei and Taehwa clearly show a diurnal cycle during KORUS-AQ. This feature is well captured by CAMS. CO at Taehwa on the other hand, exhibit a very weak diurnal cycle that is not captured by CAMS. At this site, CO in CAMS (especially ANs) shows a strong diurnal cycle. Variations of CO in the remote sites of Baengnyeong and Fukue also appears to be irregular

and episodic. Signatures of elevated CO can also be seen at these sites, some of which coinciding with pollution transport from China sampled by DC-8. The mean diurnal cycle for these five ground sites can be found in Fig. S5.

While CAMS is able to get the observed timing of $CO_2$, the modelled magnitudes of $CO_2$ (and CO) at these sites from CAMS are too high (especially for the sites in and nearby Seoul). We took the average value across a few layers near the model surface in CAMS to provide a reasonable comparison at these sites. We use model vertical layers below 95% of

the model surface pressure (i.e., if surface pressure is 1000hPa, we average the layers below 950 hPa) to account for potential weak boundary layer mixing (especially near source regions). This feature in CAMS has been discussed in section 3.3.1. Since this averaging may introduce errors in our comparison, we only evaluate CAMS in terms of relative patterns (diurnal cycle and spatial variability across sites). Note that CAMS CO and $CO_2$ along the shop tracks (to be discussed in the succeeding section) are also averaged across a few layers in the same way for consistency. We show in Fig. 9 the summary

statistics of the bias in CAMS relative to ground observations. The boxplots show that the variability of model bias in CO is in general smaller for remote sites and larger for the two sites in Seoul metropolitan. The bias in CAMS is also smaller in Fukue than in Baengnyeong, where a larger influence of pollution transport from China is observed but not well captured in CAMS. It is also worth mentioning that relative to other sites, CAMS significantly overestimates both CO and $CO_2$ at Taehwa. This may be due to the proximity of Taehwa to Seoul. The model grid spacing may not be able to resolve well the

subgrid-scale processes (emissions) and variations between Seoul and Taehwa. This overestimation is most apparent in CAMS ANs which has a coarser grid spacing (40 km for $CO_2$ and 80 km for CO) than FC16s and FC9s. In the case of $CO_2$ at Yonsei, we find lower bias in CAMS FC9s and ANs than FC16s suggesting improvements of CAMS due to better



initialization.

We take advantage of the location of the sites in Olympic Park (CO) and Yonsei University ($CO_2$) which are within Seoul metropolitan and the collocated measurements of CO and $CO_2$ in Taehwa to investigate patterns of ground-based $\Delta CO/\Delta CO_2$ in Seoul and Taehwa. Here, we only discuss observed $\Delta CO/\Delta CO_2$ since the modeled $\Delta CO/\Delta CO_2$ at these ground

sites may not be accurate given CAMS issues with vertical mixing near the surface and representativeness errors. Following similar analysis with the DC-8 $\Delta CO/\Delta CO_2$, regressions of CO to $CO_2$ at these sites can represent emission ratios of CO to $CO_2$ in Seoul metropolitan. Our estimate of $\Delta CO/\Delta CO_2$ from Olympic Park and Yonsei sites is 11.32 ppbv/ppmv. This is consistent with $\Delta CO/\Delta CO_2$ calculated from DC-8 which sampled air closely above these sites (~9 ppbv/ppmv). Our estimate of $\Delta CO/\Delta CO_2$ from the Taehwa site is 6.57 ppbv/ppmv. This is different from our DC-8 estimate of 15.3 ppbv/ppmv. Unlike

Seoul, 70% of DC-8 measurements over Taehwa are taken above 800 hPa, Over Taehwa, airborne $\Delta CO/\Delta CO_2$ varies with altitude from 8.92 ppbv/ppmv below 950 hPa, 10.28 ppbv/ppmv below 900 hPa, and 14.74 ppbv/ppmv above 400 hPa.

### 4.2 Comparison with ship observations

Two research vessels (Jangmok and Onnuri) were deployed during KORUS-OC. The two ships travelled along the Korean coast and measured CO from May 20[th] to June 5[th] (as marked in Fig. 2). Measurements of CO from ships, and biases

of CAMS FC16s, ANs, and FC9s are shown in Fig. 10. Note that CAMS values along ship tracks are also averaged across a few layers near surface in the same way CAMS at ground sites were processed. CAMS at three (out of four) ground sites tend to underestimate CO, while CAMS overestimates CO relative to ship measurements. This seems to be inconsistent with our findings with airborne measurements (i.e., CO is underestimated by CAMS at lowermost troposphere (Fig. 5 and Fig. 7). This is likely due to the differences in sampling between the airborne and ship measurements. Over sea, the DC-8 often

sampled air from China outflow while the two ships continuously sampled air over the waters regardless of the presence of China outflows. The ship measurements reflect surface conditions over waters which may also be different from what is observed by DC-8 along the vertical profile. This inconsistency is further discussed in the next section with satellite data.

### 4.3 Comparison with satellite retrievals

The total column dry air mole fractions of $CO_2$ and CO ($XCO_2$ and XCO) derived from CAMS are compared here

to $XCO_2$ from OCO-2 and GOSAT, and XCO from MOPITT and IASI. It is worth noting that satellite retrievals may have associated bias and uncertainties, which are generally larger than those of ground and airborne measurements. Slight inconsistencies also exist between MOPITT XCO and IASI XCO (George et al., 2009; 2015). We show in Fig. 11 the spatial distribution of CAMS biases against these retrievals. We also summarize the statistics in Table 4. Overall, ANs tend to agree better with satellite observations than the forecasts. For CO, CAMS XCO tends to be higher than MOPITT but lower than

IASI. In addition, CAMS XCO agrees better with MOPITT than IASI. For $CO_2$, CAMS $XCO_2$ tend to be higher than GOSAT but lower than OCO-2. FC16s, FC9s, and ANs differ from each other in terms of bias when compared to any of the four satellite retrievals although there is no clear difference in terms of RMSE. For XCO, when compared to MOPITT, ANs





are better than the two forecasts in terms of bias, RMSE, and correlation. When compared to IASI, ANs are better in terms of RMSE and correlation, but not its bias. For $XCO_2$, ANs do not show improvements from the two forecasts when compared to both OCO-2 and GOSAT retrievals. For both XCO and $XCO_2$, FC9s is not necessarily better than FC16s. In summary, ANs XCO show better agreement with satellite retrievals but this is not the case for $XCO_2$. Differences in the resolution and

amount of satellite data of XCO and $XCO_2$ could be two possible causes. The spatial and temporal resolutions of FC16s and FC9s are higher than those of ANs while ANs assimilate observational data from these satellite retrievals (except OCO-2). These two factors compete against each other. Because the size of CO data (13612 retrievals for MOPITT and 25509 for IASI over our study domain during KORUS-AQ) is much larger than that of $CO_2$ (42 for GOSAT over our domain during KORUS-AQ). This is illustrated in Fig. 10 and listed in Table 4. There are more observational constraints for CO in CAMS

resulting to better performance of ANs CO. The opposite is the case for $CO_2$. The model resolution dominates for CAMS $CO_2$ performance especially with regards to capturing spatiotemporal variability. Scatter plots of CAMS XCO and $XCO_2$ against satellite observations are also presented in Fig. S6 of the supplementary material.

We note that CAMS overestimates XCO when compared with MOPITT XCO over the West Sea (Fig. 11). This appears to be contradictory to our conclusions in section 3 and the similar inconsistency also exists when we compare

CAMS CO with ship measurements (as mentioned in Section 4.2). To further explain this inconsistency, we compare CAMS FC9s with ship measurements and satellite XCO. Because the West Sea flight group in DC-8 measurements forms a zonal 'wall' and such measurements over the West Sea are only conducted when a China outflow is expected, we separate the days when China outflows are present. The following are the days during the campaign when China outflows were expected to occur and DC-8 flights measured walls over the West Sea: May 3[rd], May 17[th], May 24[th], May 29[th], and May 30[th] On May 3[rd],

May 17[th], May 24th, and May 29[th], there are no MOPITT observations over the West Sea (Fig. S7). Therefore, the overall differences between CAMS FC9s and MOPITT observations are driven by the non-outflow days. On May 30[th], however, there are MOPITT observations over the West Sea. Unlike the overall picture (Fig. 11), we find that CAMS actually underestimates the outflows over the West Sea on that day, which is consistent with our findings in Section 3. On June 1[st] (a non-China outflow day), comparison with ship measurements indicates that CAMS FC9s overestimates CO near Korean

coast. It is also consistent with MOPITT XCO in June 1[st] (Fig. S7). This overestimation in CAMS FC9s is also captured in our comparison with Baengnyeong (highlighted by a black box in Fig. 10). We find similar overestimation using CAMS FC16s and ANs. Hence, during 'normal' conditions, CAMS tend to overestimate CO over the West Sea, whereas during China outflow events, CAMS tend to underestimate CO.

## 5 Discussions and Conclusions

We use measurements from the NASA DC-8 aircraft, five ground sites (Baengnyeong, Fukue, Olympic Park, Taehwa, and Yonsei University), and two ships (Jangmok and Onnuri) during the KORUS-AQ field campaign, along with four sets of satellite retrievals (MOPITT XCO, IASI XCO, OCO-2 $XCO_2$, and GOSAT $XCO_2$) to evaluate the capability of a high-resolution global modeling system (CAMS) in simulating anthropogenic combustion. Specifically, we evaluate the





performance of CAMS FC16s, FC9s, and ANs of $CO_2$, CO, and their relationships. Our assessment of the overall performance of CAMS against DC-8 measurements show that: (1) The nominal background $CO_2$ in CAMS is slightly overestimated (bias is 2.2 ppmv for FC16s, 0.7 ppmv for FC9s, and 0.3 ppmv for ANs), which is further improved by $CO_2$ analysis. The overall overestimation of $CO_2$ might be associated with the biogenic bias correction. On the other hand, CO is

generally underestimated by CAMS (bias is -19.2 ppbv for FC16s, -16.7 ppbv for FC9s, and -20.7 ppbv for ANs); and (2) Among the three forecasts/analysis configurations, FC9s are more accurate and consistent overall than FC16s and ANs because of the finer model resolution and improved initialization. While ANs are coarser in resolution, they generally perform better than FC16s as the impact of initialization surpasses the impact of resolution (Fig. S2). We also classify the airborne measurements into five groups based on land cover below the flight tracks and associated pollution sources. While

$CO_2$, CO, and their relationships vary across these five groups, CAMS perform well in terms of simulating regional pattern of anthropogenic combustion. This is because: 1) CAMS simulations of both species have relatively low bias; and 2) CAMS reproduces $\Delta CO/\Delta CO_2$ observed by DC-8. Both CAMS and DC-8 show more efficient combustion (low $\Delta CO/\Delta CO_2$) over Seoul than over the West Sea which is representative of Chinese outflows. Our case study on the May 24th flight over the West Sea indicates that the Chinese outflow is captured by CAMS. However, the modeled CO and $CO_2$ concentrations are

significantly underestimated (by -2 to -4 ppmv for $CO_2$ and -86 to -88 ppbv) especially within the lowermost troposphere. This suggests that, although CAMS emission ratios are relatively consistent with $\Delta CO/\Delta CO_2$, the absolute magnitude of China emissions are still underestimated. CAMS also show poorer performance at local-to-urban scales as exemplified by our case study in the June 4th flight where larger variations near point sources were not represented in CAMS. Our comparisons with measurements from ground sites and two ships indicate that: (1) the diurnal cycle of CO and $CO_2$ are

stronger over urban environments and such periodic features are reasonably captured by CAMS; (2) vertical mixing near sources (such as Seoul) is too weak in CAMS and needs to be improved; and (3) in some cases, FC9s do not show improvements from FC16s (such as over Seoul and the point sources during the June 4th flight), implying large spatiotemporal errors in emission inventories. In these cases, increasing the spatiotemporal resolution might even weaken the simulation results, whereas lower resolution usually agrees better with observations as it "diffuses" the error of the

emissions. We also compared XCO and $XCO_2$ derived from CAMS to satellite retrievals from four instruments (MOPITT CO, IASI CO, OCO-2 $CO_2$, and GOSAT $CO_2$). We find that ANs XCO show better agreement with satellite retrievals compared to the forecasts, while ANs $CO_2$ is no better than the forecasts. We attribute this contrast to significant differences in the number of XCO and $XCO_2$ satellite data potentially available for assimilation.

We recognize the following limitations of this work. (1) The temporal distribution of airborne measurements are not

completely independent from their spatial distributions. For example, most of the measurements in the West Sea group are conducted before noon, whereas measurements over Seoul-Busan jetway are concentrated in the afternoon. (2) CAMS is only evaluated over South Korean peninsula and surrounding waters during the campaign (May 1st to June 10th). More work is needed to determine if our findings are valid over other regions. For example, Agusti-Panareda et al. (2014) reported the overall overestimation of $CO_2$ in spring over the whole Northern Hemisphere and it is associated with biogenic flux



correction. (3) Inconsistencies exist even among different satellite products (George et al., 2009; 2015), thus limiting our comparisons with CAMS to relative differences; and 4) Our comparison of CAMS with ground and ship measurements are only qualitative and indicative as CAMS surface concentrations are significantly higher than surface observations and not comparable.

Finally, this study has important implications on the design and implementation of current and future prediction system for atmospheric composition and air quality. Although CAMS captured the regional combustion signatures, it still has difficulty representing the variability at local-to-urban scales even at finer resolution. This suggests both improvements in observational constraints and model representation of relevant processes (e.g., emissions and boundary layer mixing).

***Data availability.*** CAMS 16-km forecasts, and analyses are available online (http://apps.ecmwf.int/datasets/data/cams-nrealtime/levtype=sfc/). CAMS 9-km forecasts are available upon request. Observational data from KORUS-AQ will be open to public soon (https://www.air.larc.nasa.gov/cgi-bin/ArcView/korusaq). All the satellite data used in this study are

available online. MOPITT CO and OCO-2 $CO_2$ can be downloaded at https://reverb.echo.nasa.gov/reverb/. IASI CO can be found at http://ether.ipsl.jussieu.fr/ether/pubipsl/iasi_CO_uk.jsp. GOSAT $CO_2$ data after 2014 is available at http://www.gosat.nies.go.jp/en/.

***Acknowledgements.*** This work is supported by NASA KORUSAQ NNX16AE16G. We thank the KORUS-AQ team for

observational data, the CAMS global production team for the model products of CO and $CO_2$, MOPITT, IASI, OCO-2, and GOSAT data teams for satellite data. IASI CO is provided by LATMOS/CNRS and ULB. We acknowledge NASA and the OCO-2 project for OCO-2 $CO_2$ data. The authors thank Dr. Cenlin He for helpful comments on improving the paper. The CAMS data was generated using Copernicus Atmosphere Monitoring Service Information [2016].

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




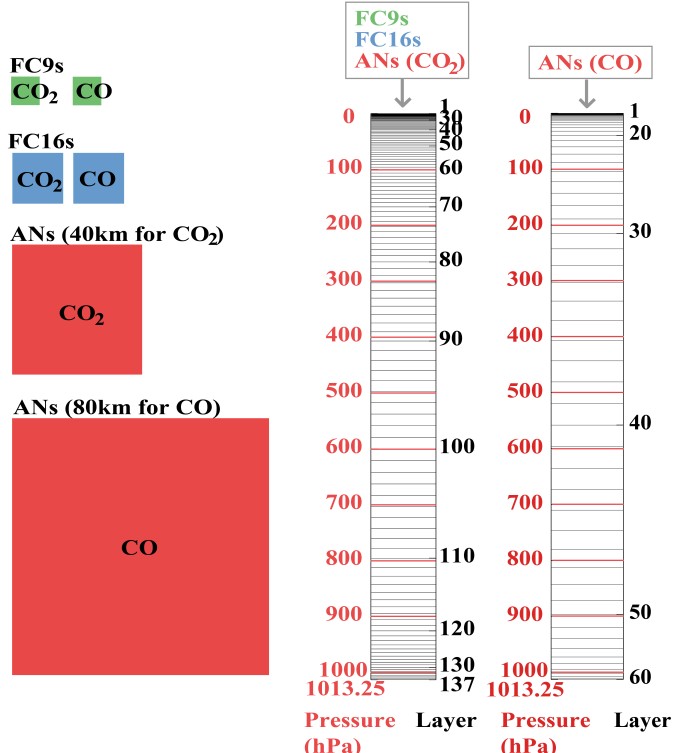

**Figure 1.** Model grid sizes of the CAMS and vertical structures of the model layers assuming the surface pressure being 1013.25hPa. FC9s, FC16s, and ANs for $CO_2$ (40 km) have 137 vertical layers. ANs for CO (80 km) have 60 vertical layers.





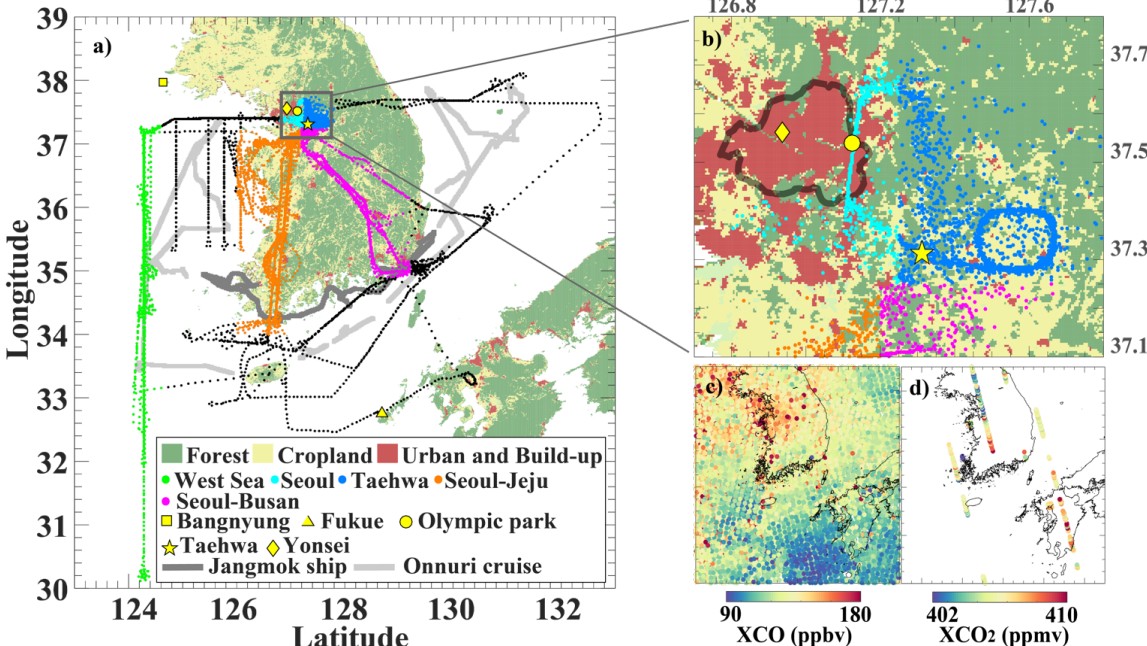

**Figure 2.** Domain of the study and KORUS-AQ measurements used in this study. Panel (a) shows land cover of the domain (Broxton et al., 2014), DC-8 aircraft tracks, ship tracks, and location of ground sites. The airborne measurements are classified into 5 groups (West Sea, Seoul, Taehwa, Seoul-Jeju jetway, and Seoul-Busan jetway), as marked in luminous green, luminous blue, mazarine blue, orange, and

5  magenta. The ground sites are labelled with luminous yellow markers. Olympic park and Yonsei sites are located in urban regions (Seoul) while Baengnyeong and Fukue site are located in remote regions. Taehwa site is located in a forest nearby Seoul. Tracks of the two ships are marked in dark grey (Jangmok ship) and light grey (Onnuri ship). Also shown in (b) is the zoomed-in version of the grey box in panel (a). Panel (c) shows a composite MOPITT XCO retrievals during KORUS-AQ campaign while panel (d) shows OCO-2 XCO$_2$ retrievals in the same time period.





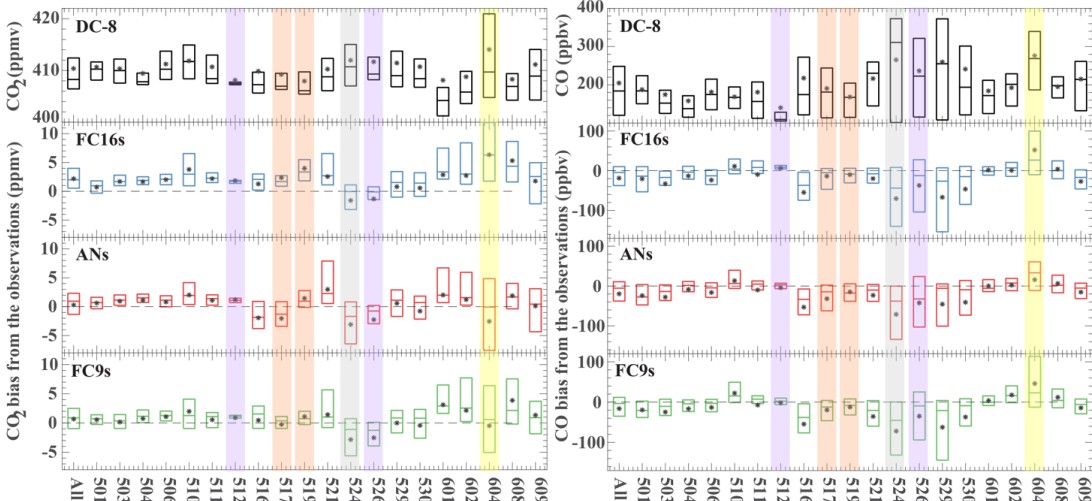

**Figure 3.** Boxplot for each individual flight. The flight date (MDD) for each boxplot is indicated in the bottom x-axis. Note that the dates here are in UTC time instead of Korea time. The left panel is for $CO_2$ and the right panel is for CO. The first row corresponds to the boxplot of the abundances measured by DC-8 aircraft. The second, third, and fourth rows correspond to the boxplot of the bias of FC16s, ANs, and FC9s relative to DC-8, respectively. The purple shade marks the flights with frontal passage, and orange shade marks the flights that may possibly be affected by biomass burning. The grey shade marks the flight measuring China outflow while yellow shade marks the flight surveying point emission sources.



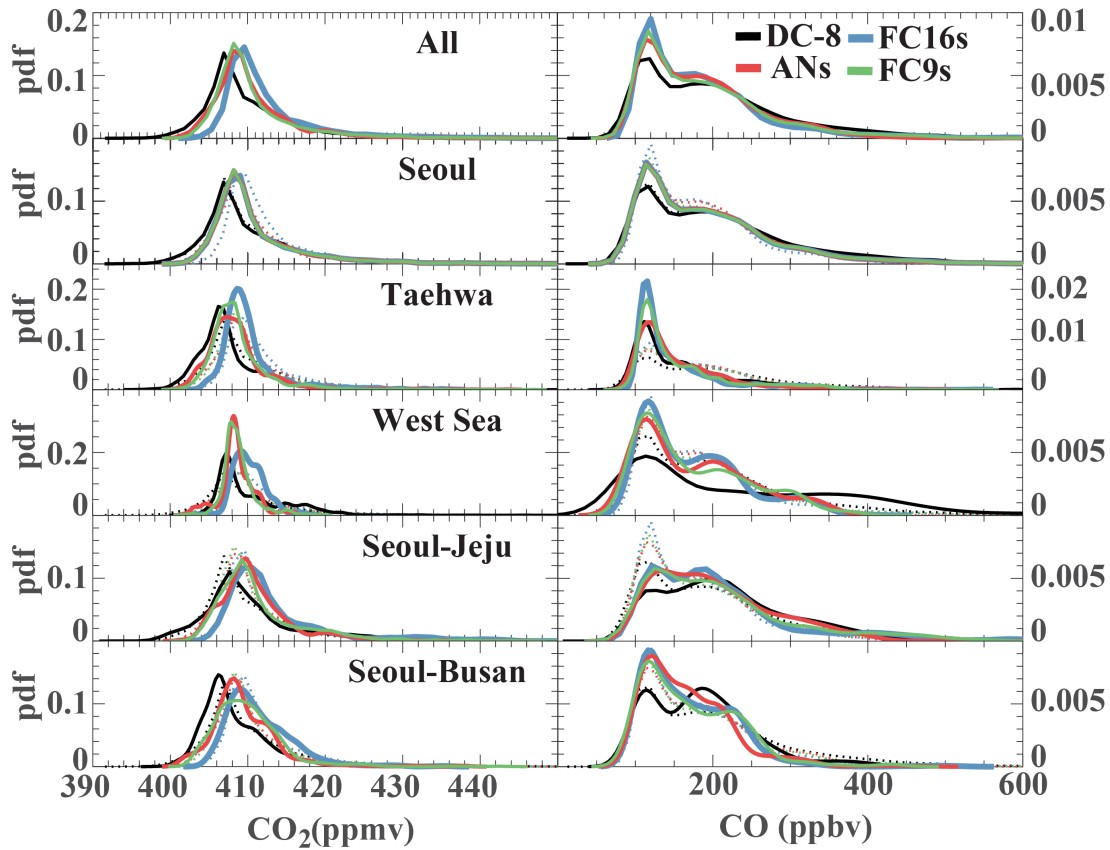

**Figure 4**. Probability density functions (pdfs) of $CO_2$ and CO for each flight group. Solid lines are pdfs for each group while the dashed lines are pdfs for all groups.



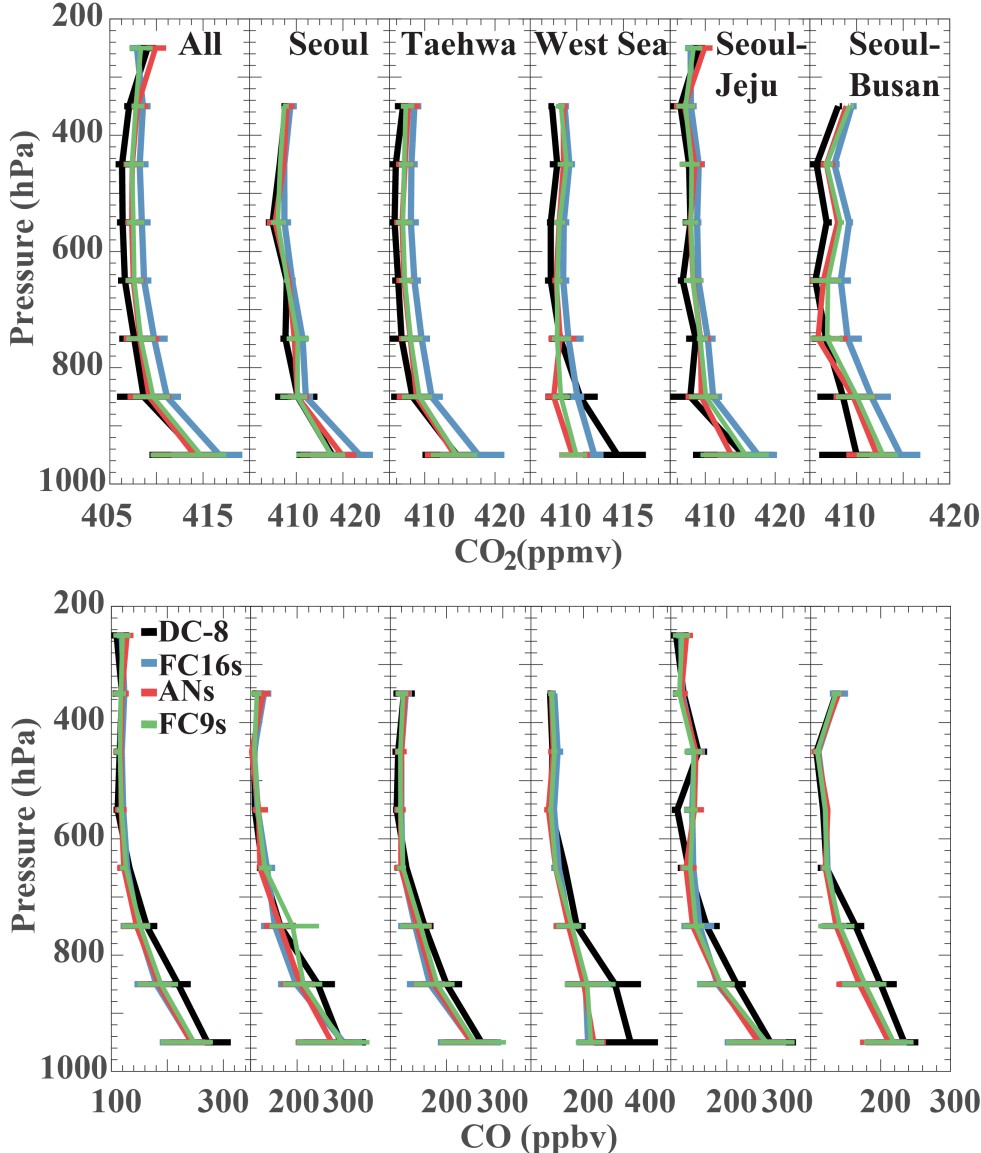

**Figure 5.** Averaged vertical profiles of $CO_2$ and CO mixing ratios from DC-8 and CAMS for each flight group. Horizontal bars correspond to the interquartile ranges (between 25th and 75th percentiles) of the layer bin.




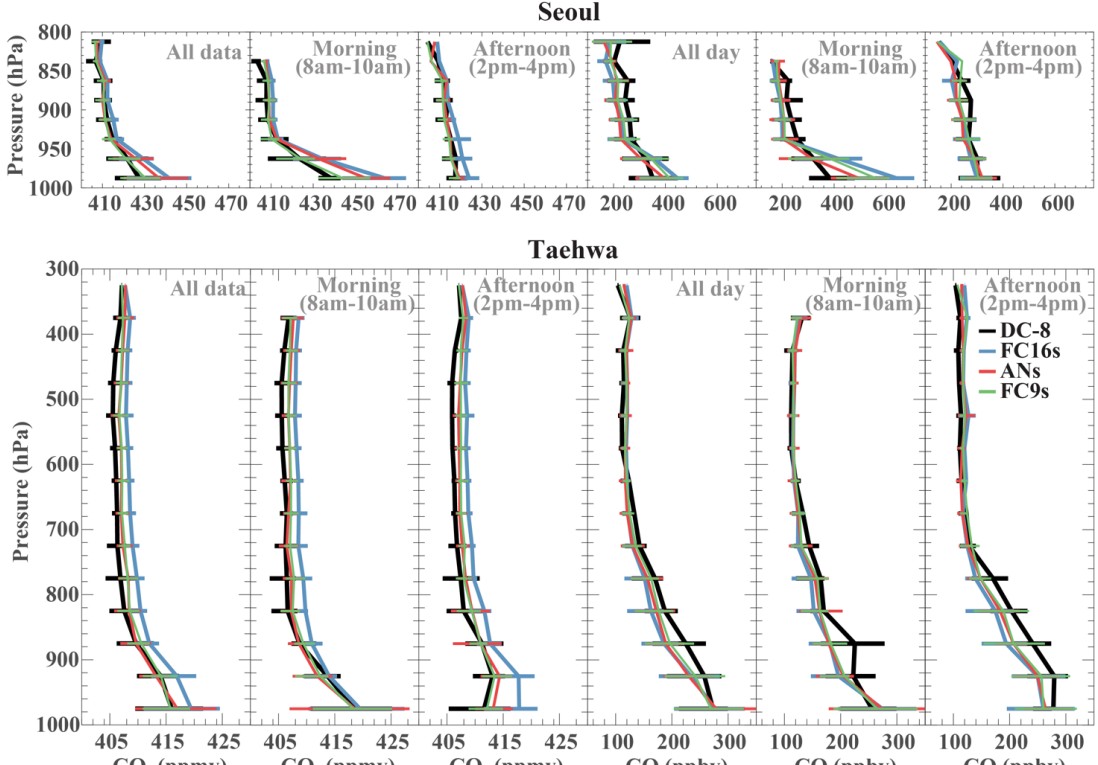

**Figure 6.** Temporal variation of averaged vertical profiles of $CO_2$ and CO mixing ratios from DC-8 and CAMS over Seoul and Taehwa
flight groups. The first, second, and third columns are averaged $CO_2$ profiles for all day, morning (8-10am), and afternoon (2-4pm),
respectively. Horizontal bars correspond to interquartile ranges (between 25th and 75th percentiles) of the profiles. The fourth, fifth, and
sixth column are the same as the first three columns but for CO.



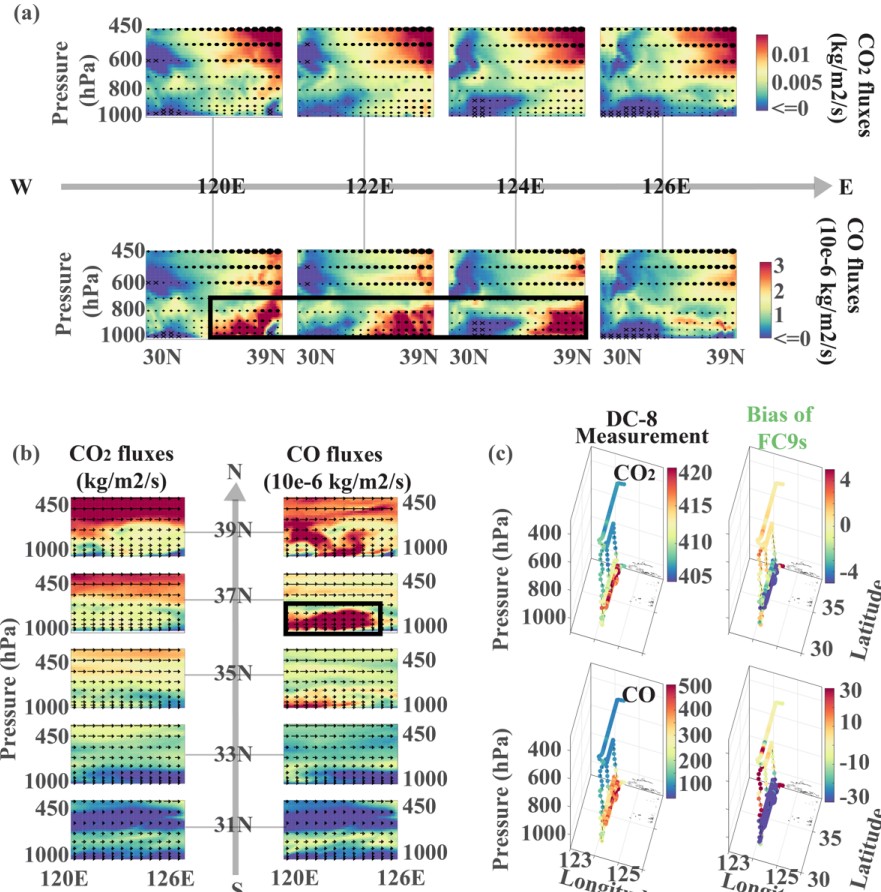

**Figure 7.** Case study for the flight on May 24[th] (UTC time). (a) Vertical distributions (hereafter denoted as 'sections') of fluxes (kg/m[2]/s) at 9:00 am on May 25[th] (Korea time) in meridional direction. Dots represent meridional winds going from west to east (i.e., from China to Korea) and crosses represent meridional winds with the opposite direction. Sizes of the dots and crosses are proportional to the wind speed. 'Sections' on the top are for $CO_2$ fluxes and the bottom are for CO fluxes. (b) 'Sections' of fluxes (kg/m2/s) at 9:00 am on May 25[th] (Korea time) in zonal direction. Arrows represent meridional winds. 'Sections' in panel (b) share the same colorbar as panel (a). (c) DC-8 aircraft measurements (left column) and bias of CAMS along the flight track over West Sea (right column). The top row is for $CO_2$ and bottom row is for CO.





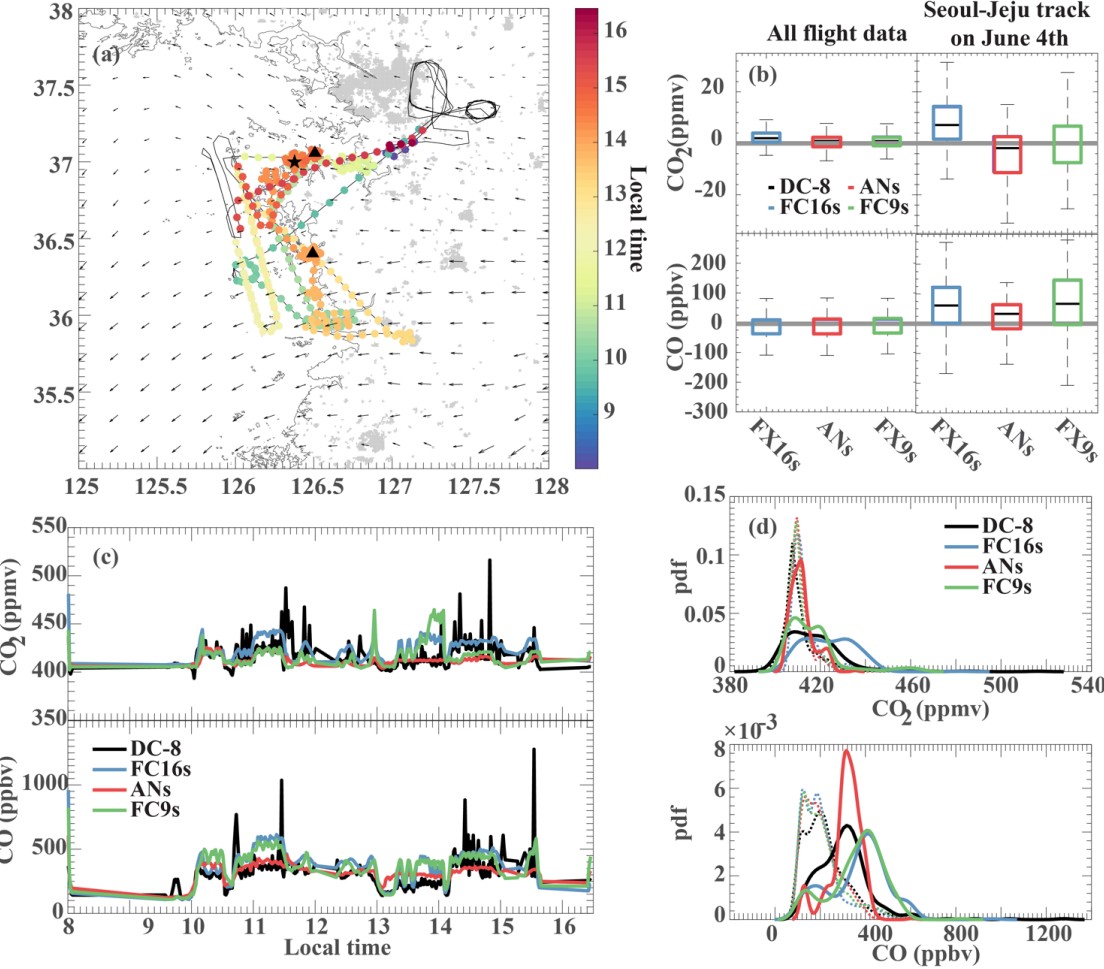

**Figure 8.** Case study for the flight on June 4[th] (UTC time). (a) Flight track of DC-8 aircraft in the Seoul-Jeju jetway group for this day. The Daesan chemical facility is marked as black pentagram and two power plants are marked as black triangles. Arrows correspond to 950 hPa wind field at 12:00pm local time. (b) Boxplot of CAMS bias from all the DC-8 aircraft measurements during the campaign (left), and from measurements on June 4[th] in the Seoul-Jeju jetway group (right). Top row is for $CO_2$ and bottom row is for CO. (c) Time series of DC-8 aircraft measurements and CAMS during the flight. (d) pdfs of CO and $CO_2$ for measurements on June 4[th] of the Seoul-Jeju jetway group (solid) and for all groups (dashed).





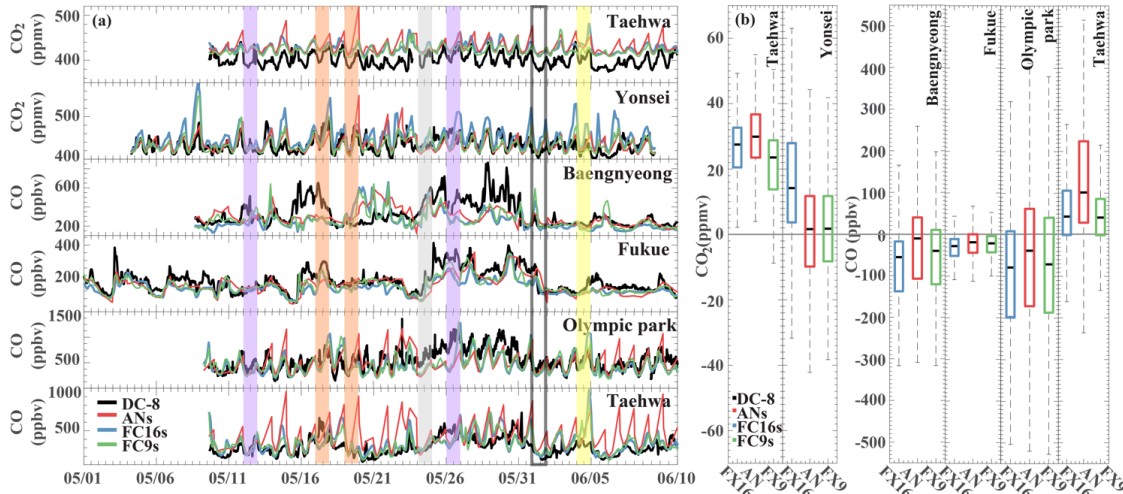

**Figure 9.** Comparisons of CAMS against ground site measurements. Values of CAMS are averages across layers with pressure higher than 95% of the surface pressure. (a) Time series of measured and CAMS $CO_2$ from the Taehwa and Yonsei sites, and CO from the Bangnyung, Fukue, Olympic park, and Taehwa sites. Shades denote same events as they do in Fig. 3. (b) Boxplot of CAMS bias for $CO_2$ at the Taehwa and Yonsei site measurements, and for CO at the Bangnyung, Fukue, Olympic park, and Taehwa sites.





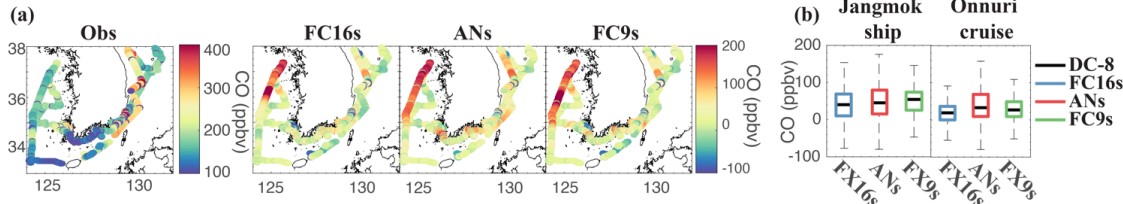

**Figure 10.** Comparisons of CAMS CO against ship measurements. Values of CAMS are averages across layers with pressure higher than 95% of the surface pressure. (a) Bias of CAMS CO against ship measurements along the ship track. (b) Boxplot of CAMS bias for CO compared with ship measurements.





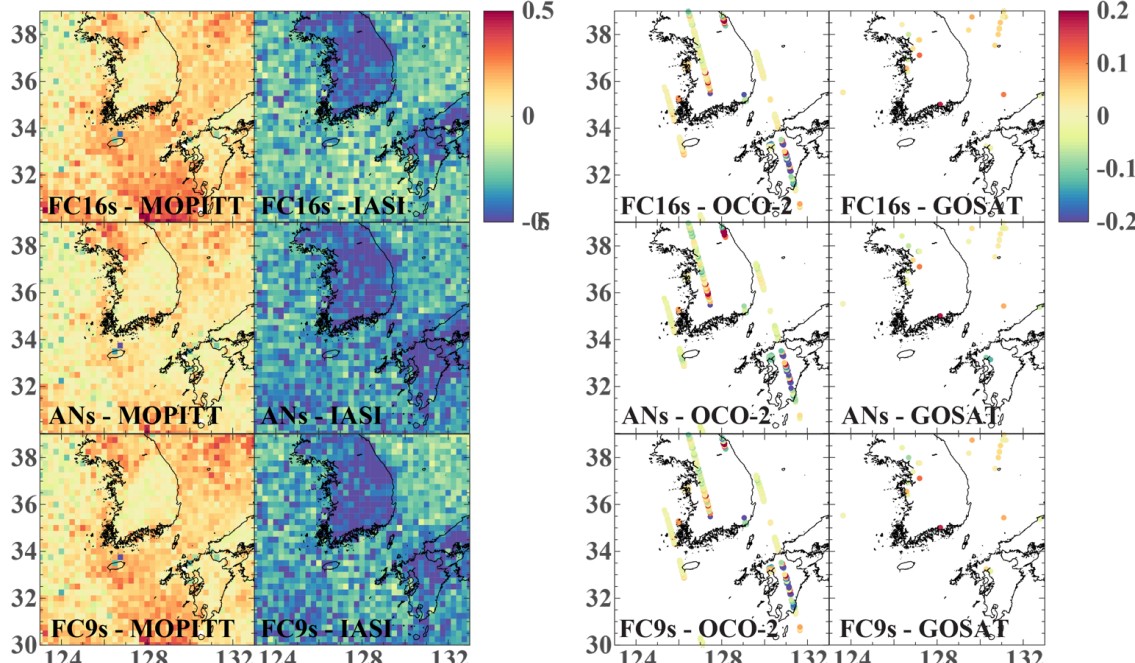

**Figure 11.** Spatial distributions of CAMS bias against satellite retrievals. For XCO, the unit is $10^{18}$ molecules/cm$^2$ while for XCO$_2$, the unit is $10^{21}$ molecules/cm$^2$.



**Table1.** Configuration of CAMS global atmospheric composition products valid during the period of the KORUS-AQ Field Campaign
(May to June 2016). The tracers evaluated in this paper are highlighted in bold face. Time availability is in number of days with respect to
real time (N/A is used when this is not applicable).

| CAMS product | Atmospheric composition tracers | Horizontal resolution | Number vertical levels | Initial conditions: Meteorology | Initial conditions: Atmospheric composition | Time availability observations/analysis of atmospheric composition | Time availability of product |
|---|---|---|---|---|---|---|---|
| **AN_CHEM** | Reactive gases (**CO**,O3,NO$_2$,etc) and aerosols | 80 km | L60 | Own analysis | Own analysis | <1day | <1day |
| FC_CHEM | Reactive gases (**CO**,O3,NO$_2$,etc) and aerosols | 80 km | L60 | AN_CHEM | AN_CHEM | <1day | 0 days (real time) |
| **AN_GHG** | **CO$_2$**, CH$_4$ | 40 km | L137 | Own analysis | Own analysis | 2-4 days | 4 days |
| **FC16s** | CO$_2$, CH$_4$ and **linCO** | 16 km | L137 | ECMWF operational analysis | Previous 1-day forecast | N/A | 1 day |
| **FC9s** | **CO$_2$**, CH$_4$, **linCO** and tagged tracers | 9 km | L137 | ECMWF operational analysis | AN_GHG 4-day fc for CO$_2$/CH$_4$ and AN_CHEM for linCO | 4 day for AN_GHG; <1day for AN_CHEM | 1 day |



**Table 2.** Measurements during KORUS-AQ.

| | | | $CO_2$ | CO |
|---|---|---|---|---|
| Airborne measurements | NASA DC-8 aircraft | Instrument | LI-COR | DACOM |
| | | Time Response | 1 second | 1 second |
| | | Precision | < 0.1 ppmv | < 1% or 0.1 ppbv |
| | | Accuracy | 0.25 ppmv (Vay et al., 2003) | 2% (Warner et al., 2010) |
| Ground site measurements | Baengnyeong (37.97N,124.63E) | Instrument | / | Teledyne Gas analyzer |
| | | Data intervals | / | 1 hour |
| | Fukue (32.75N,128.68E) | Instrument | / | 48C |
| | | Data intervals | / | 1 hour |
| | Olympic Park (37.52N,127.12E) | Instrument | / | KENTEK CO analyzer |
| | | Data intervals | / | 5 minutes |
| | Taehwa (37.31N,127.31E) | Instrument | LI-COR LI-7500 | Thermo 48i |
| | | Data intervals | 1 hour | 1 hour |
| | Yonsei (37.56N, 126.94E) | Instrument | G2201-I $CO_2/CH_4$ carbon stable isotope analyzer | / |
| | | Data intervals | 30 minutes | / |
| Ship measurements | R/V Jangmok | Instrument | / | Thermo 48i-TLE |
| | | Data intervals | / | 1 minute |
| | R/V Onnuri | Instrument | / | Thermo Scientific, Inc., Model 48C |
| | | Data intervals | / | 1 minute |
| Satellite measurements | | | OCO-2 | / |
| | OCO-2 | Date product | Level 2 v7 Full Product $XCO_2$ | / |
| | | Resolution | 2.25x1.29-km Global coverage ~16 days | / |
| | | Revisit time | 1:18 - 1:33 pm | / |
| | | Uncertainty | 1-2 ppm $XCO_2$ (Boesch et al., 2011 and references therein) | / |
| | GOSAT | Date product | Level 2 V02 | / |




| | Resolution | 10.5 x 10.5 km | / |
|---|---|---|---|
| | | ~12 days | |
| | Revisit time | ~1:00 pm | / |
| | Uncertainty | 2 ppm for retrieval errors of | / |
| | | XCO$_2$ | |
| | | Griffith et al. 2011; | |
| | | Crisp et al. 2012 | |
| MOPITT | Date product | / | TIR/NIR Level 2 v6 XCO |
| | Resolution | / | 22 x 22 km |
| | | | ~3-4 days |
| | Revisit time | / | 10:30 am |
| | Uncertainty | / | 0.09e18 molecules/cm$^2$ for |
| | | | total column retrieval; |
| | | | (Deeter et al., 2014) |
| IASI | Date product | / | Level 2 FORLI XCO |
| | Resolution | / | 12 km x 12 km |
| | | | twice a day |
| | Revisit time | / | |
| | Uncertainty | / | <13% for FORLI (Wachter |
| | | | et al., 2012) |



**Table 3.** Statistics of CAMS performance evaluated against satellite observations.

| | | Seoul | Taehwa | West Sea | Seoul-Jeju | Seoul-Busan | All |
|---|---|---|---|---|---|---|---|
| $\Delta CO/\Delta CO_2$ (ppbv/ppmv) | DC-8 measurement | 9.09±0.48 | 15.3±0.56 | 28.17±0.75 | 10.37±0.31 | 15.86±0.73 | 13.29±0.21 |
| | FC16s | 9.84±0.29 | 14.31±0.40 | 30.86±1.64 | 13.00±0.27 | 13.39±0.51 | 12.28±0.15 |
| | ANs | 8.21±0.45 | 13.71±0.48 | 30.60±1.73 | 14.98±0.45 | 12.68±0.47 | 12.60±0.2 |
| | FC9s | 11.56±0.62 | 16.06±0.57 | 32.44±1.77 | 11.68±0.35 | 13.87±0.54 | 12.52±0.2 |
| Correlation of CO and $CO_2$ | DC-8 measurement | 0.78 | 0.68 | 0.89 | 0.62 | 0.60 | 0.66 |
| | FC16s | 0.94 | 0.83 | 0.42 | 0.83 | 0.74 | 0.82 |
| | ANs | 0.77 | 0.71 | 0.25 | 0.61 | 0.76 | 0.63 |
| | FC9s | 0.78 | 0.70 | 0.36 | 0.60 | 0.73 | 0.65 |
| Correlation of $Bias_{CO}$ and $Bias_{CO_2}$ | FC16s | 0.90 | 0.61 | 0.80 | 0.46 | 0.55 | 0.61 |
| | ANs | 0.66 | 0.59 | 0.82 | 0.36 | 0.63 | 0.51 |
| | FC9s | 0.64 | 0.52 | 0.82 | 0.33 | 0.54 | 0.49 |



**Table 4.** Statistics of CAMS performance compared against satellite observations.

| | | CO | | CO$_2$ | |
|---|---|---|---|---|---|
| | | MOPITT | IASI | OCO-2 | GOSAT |
| total observations during campaign | | 13612 | 25509 | 4591 | 42 |
| bias (molecules/cm$^2$) | FC16s | -1.13E+17 | 8.28E+16 | 9.30E+18 | -2.64E+19 |
| | ANs | -6.42E+16 | 1.36E+17 | 4.48E+19 | 1.05E+19 |
| | FC9s | -1.01E+17 | 7.52E+16 | -1.31E+19 | -1.28E+19 |
| RMSE (molecules/cm$^2$) | FC16s | 2.47E+17 | 4.19E+17 | 7.11E+19 | 5.67E+19 |
| | ANs | 2.31E+17 | 4.12E+17 | 8.48E+19 | 6.42E+19 |
| | FC9s | 2.56E+17 | 4.19E+17 | 8.29E+19 | 5.49E+19 |
| correlation | FC16s | 0.65 | 0.44 | 0.88 | 0.78 |
| | ANs | 0.66 | 0.52 | 0.85 | 0.63 |
| | FC9s | 0.61 | 0.45 | 0.85 | 0.75 |