# Peer review of "Evaluating High-Resolution Forecasts of Atmospheric CO and CO2 from a Global Prediction System during KORUS-AQ Field Campaign"

_Atmospheric Chemistry and Physics, 2018_

## Referee Comment (RC1) · Anonymous Referee #1 · 23 Apr 2018

The authors have presented an evaluation of the CAMS prediction system, focusing on CO and CO2, during the KORUS-AQ campaign. They evaluated three different CO and CO2 forecast and analysis products: 16-km CO and CO2 forecasts, 9-km CO and CO2 forecasts, and analyses of CO and CO2 at 80 km and 40 km, respectively. The CAMS products were compared to the KORUS-AQ aircraft data as well as to ground-based and satellite measurements of CO and CO2. They found that CAM overestimated CO2, suggesting a positive bias in background CO2, whereas it underestimated CO, with the underestimate confined mainly to the lower troposphere. The authors also found that CAMS underestimates the outflow of pollution from China, possibly due to an underestimate of Chinese emissions. The study is a nice evaluation of CAMS CO

and CO2 under unique conditions. I have no major concerns about the analysis. My main concern is about the appropriateness of the manuscript for ACP. As a model evaluation study, I think it is better suited for GMD than ACP. My comments below are relatively minor, but must be address before the manuscript can be accepted for publication, if the Editor decides it is suitable for ACP.

Comments

1. There is no mention of the CAMS OH field, which is critical for the simulation of CO. What is the global mean OH from the analyses and forecasts? On page 5, lines 28, it is mentioned that the 16-km CO forecasts use a linear chemistry scheme. A brief description of the scheme, either in the manuscript or in the supplement, would be helpful.

2. It is stated that the overestimate in CO2 is associated with the bias correction in the biogenic source of CO2, but there is no discussion of this "bias correction". Furthermore, in Agusti-Panareda et al. (2016) CAMS CO2 was underestimating CO2 observations from the surface in situ network and from TCCON, which the "bias correction" (the biospheric flux adjustment) reduced. Why is CAMS overestimating CO2 here? A discussion is needed about the treatment of the biospheric fluxes in CAMS and its possible impact on the modeled CO2 over Korea.

3. On page 11 it was shown that the model produced steeper vertical gradients in CO and CO2 than observed over Seoul, which the authors suggested may be due to weak boundary layer mixing. Since CAMS seems to perform better over Taehwa, it would be interesting to compare the vertical gradients over Seoul and Taehwa in CAMS and in the observations to see if the issue is mainly an inability of CAMS to capture the PBL heights over the Seoul urban environment.

4. I am surprised that the analyses are not much better than the forecasts. Indeed, it seems as though the 9-km forecast is better than the analyses in some cases. I think it would be helpful for the reader if the authors expanded the description of the analyses

to give the reader more information about the configure and quality of the analyses. Figure S1 and the brief text on page 5 are not enough.

5. The discussion of enhancement ratios is confusing. It is unclear if the authors are using the slope of the $CO/CO_2$ relationship or the slope of delta CO/delta $CO_2$ relationship. The two approaches are different. The description in the text suggests that they are using the RMA regression of $CO/CO_2$ to assess the combustion sources, but throughout the text there is use of the delta CO/delta $CO_2$ notation. If they are indeed calculating an enhancement ratio (delta CO/delta $CO_2$) above the background, how is the background being calculated? How sensitive is the analysis to the definition of the background?

6. The authors found that CAMS underestimated CO during China outflow events, but overestimated it under normal conditions. What are the different source regions for air reach the West Sea during "outflow" and "normal" conditions? To what degree is the model bias due to CAMS not capturing this difference in transport as compared to it not have the correct balance of emissions in China?

Technical Comments

1. Page 4, line 26: add a comma between "forecast" and "CO2".

2. Page 4, line 28: add "the" between "on" and "free-running".

3. Page 5, line 1: Is Figure 1 really necessary? I don't think it adds much to the manuscript. Since there are already 11 figures, I would suggest removing Figure 1.

4. Page 6, lines 1 and 2; add "the" before "South Korean peninsula".

5. Page 6, line 7: remove "including" before the list of the three questions.

6. Page 6, line 8: The wording for question (3) needs improving. The English is not quite correct.

7. Page 8, lines 22 and 23: The revisit and overpass times seem to be used interchangeably here. The revisit time of OCO-2, for example, is 16 days since it is in the A-Train orbit. However, the local overpass time is around 1:30 pm. For GOSAT the revisit time is 3 days.

8. Page 8, line 26: Change 0.09e18 from e-notation to standard SI notation.

9. Page 9, line 5: Please add "is" between "CO2" and "associated".

10. Page 10, line 1: The variance in CO in the May 3rd data does not seem larger than average to me. In fact, it seems to be smaller than average.

11. Page 10, lines 12 and 14: Please change "tale" to "tail".

12. Page 12, line 4: It is unclear what is meant by the statement that "the wind speeds dominate the transport flux variations in CO2." It the argument here that the meteorological uncertainty is the dominant contribution to the uncertainty in the forecast and analysis fields? If so, how does one come to that conclusion from Figure 7?

13. Page 15, line 23: Section 4.2 only discusses the comparison to ship data of CO, not CO and CO2.

14. Page 15, line 23: change "shop tracks" to "ship tracks".

15. Page 17, line 7: Change "size of CO data" to "amount of CO data".

16. Page 40, Table 3: The title for the table is wrong. This is the same title as for Table 4.

17. Figure S1: Should the labels "FX9s" and "FX16s" in the figure be "FC9s" and "FC16s"?

---

## Referee Comment (RC2) · Anonymous Referee #2 · 22 May 2018

This manuscript presents a comparison between three modelling products of the Copernicus Atmosphere Monitoring Service (CAMS) and in situ measurements from the KORUS-AQ (and KORUS-OC) campaign in the vicinity of the Korean peninsula during May/June 2016. Airborne, surface, ship-based, and satellite measurements of CO and CO2 are compared to the CAMS analysis and two forecast products at different spatial resolutions. The statistical analysis is relatively straightforward and clearly laid out, and some patterns of over- and underestimation are found for the two tracers under different conditions. The importance of vertical transport in the understanding of these differences could be further explored, as outlined below. While I understand the other reviewer's comment that this manuscript might be a better fit for GMD, as it

is assessing the capabilities of a specific modelling system, the general conclusions about the potential underestimation of CO emissions from China make it relevant for a broader audience as well. This is ultimately an editorial decision. However the quality of the manuscript, datasets and analysis is good, and appropriate for publication. Below are some suggestions, some major, some minor, on how the analysis might be slightly extended in order to better understand the processes driving the model-data mismatch.

Section 3.3: In the discussion about the relative agreement in the profile for CO2 vs. the disagreement between the lower atmosphere values between the observations and the model, a discussion of the relevance of the mixing height and/or planetary boundary layer was somewhat lacking. A difference in profile shape can be attributed to incorrect fluxes, incorrect mixing, or a combination of the two. By having two tracers with differing results, it should be possible to deepen this analysis a bit. There is further discussion about the vertical gradients of the tracers, but no attempt is made to diagnose the PBL height. Given the model data and the meteorological information from the aircraft profiles, this should be possible. Could you at least comment on this, and why such an approach was not undertaken? It is even suggested that there might be a "possible weaker boundary layer mixing in CAMS". Here diagnosing the PBL height (as a function of time) from both the model fields and the profiles might be enlightening.

Another interesting point might be the representation of urban effects for Seoul in particular. Here it would be interesting to compare the PBL height as modelled vs. measured in the vicinity of Seoul compared to other less rural sites. However this may be beyond the scope of this study.

For the special case of Seoul, the low altitude measurements were taken during missed approaches at the airport. Given all the air traffic in the region, might it be that the CO in this area is locally very much enhanced, and as such not representative of even the relatively small spatial footprint of the CAMS model? Here perhaps a referral to a

relevant paper by Boschetti et al. (Tellus B, 2015) looking at enhancements of CO in the boundary layer from commercial airline measurements might be relevant.

Regarding the assessment of the outflow over the West Sea, I was confused by the phrase: "Hence, the wind speeds dominate the transport flux variations in CO2." I'm not sure what is meant here. Is this because the outflow pattern wasn't as strong as for CO? But aren't both flux variations (more or less) linearly dependent on wind speed anyhow? Please clarify.

The discussion about the correlation of CO and CO2 over the West Sea is quite interesting, and invites further inquiry. The suggestion that the difference in time factors for anthropogenic CO and CO2 (with the former having constant monthly values and the latter having diurnal variability) should effect the correlation over Korea as well. Could it be explained by the differences in transport times, e.g. diurnal CO2 emissions peak in daytime while measurements are being made over Korea, whereas daytime measurements over the West Sea represent nighttime emissions from China, where the difference in time factors is at a maximum? In terms of just the correlation in the fluxes, it should be easy to test if EDGAR has a higher spatial correlation between CO2 and CO in Korea vs. China.

The analysis of the satellite data is not particularly illuminating, with the exception of the separation of MOPITT data into those influence by outflow. Regarding the use of the OCO-2 data, most of the data references are pre-launch, and should be updated. Wunch et al. 2017 would be a better up to date reference than those from 2011, and an updated estimate of the OCO-2 precision, even if it is coming from grey literature (such as the ACOS OCO-2 User's Guide) would be preferable to a largely theoretical assessment from Boesch et al., 2011. It is unclear what is meant by the "recommended quality control" in section 2.2.4. Does this mean the standard quality flag? Was the bias correction applied? Was a certain warn-level threshold used? Please elaborate.

If the Taylor skill score is being used for the assessment of the forecasting skill as in

section 3.1, the equation should be in the main paper, and not just in the suppplement. Please include it here as well.

P4, L18-20: The text here states that the CO analysis runs at "approximately 40 km horizontal resolution", but in Figure 1 it is shown to be 80 km horizontal resolution. Later on page 5 80 km is given again, and the text on P4 refers to that fact that the CO2 analysis is at a higher spatial resolution (in both the horizontal and vertical). Please ensure that the information is consistent and correct.

Minor/typographical comments:

P2, L11: show -> shows

P2, L12: "over Seoul metropolitan" -> either "over the Seoul metropolitan area" or "over Seoul"

P3, L6: near-real time -> near-real-time

P3, L16: field -> field campaign

P4, L20: Perhaps this should be one sentence?

P4, L25: 4-days shouldn't be hyphenated (four days)

P4, L26: 16km -> 16 km

P5, L26: The -> the

P6, L1, L2, and often afterwards: South Korean peninsula -> the South Korean peninsula

P6, L8-10: The third scientific question needs to be restated. It doesn't make sense as it is written here.

P6, L27: data is -> data are

P6, L30-31: Wouldn't UTC be one day behind local time?

P7, L7: combustion signatureS (plural needed to match grammar)

P7, L10-12: Not sure which preposition should be used to describe the jetway flights, I would suggest "in", but consistency is more important. Also check the grammar: "Flights in the Seoul-Busan jetway were designed to capture... Flight in the Seoul-Jeju jetway, on the other hand, sampled air over..."

P7, L17: Baengnyeong site is located in less populated Baengnyeong Island, Incheon which is northwest of Seoul. -> The Baengnyeong site is located on the sparsely populated Baengnyeong Island, Incheon, northwest of Seoul.

P7, L19: "on remote" -> "on the remote"

P8, L21: resolutions -> the resolutions

P9, L2: Here is the first of many instances of referring to the in situ measurements collected from the DC-8 aircraft as simply "DC-8". As a reader I found this jarring. Perhaps instead refer to the dataset as the "DC-8 in situ data" or the "aircraft data" or "the airborne measurements"?

P9, L8: inconsistent description of correlation range (to vs. -)

P9, L12: CAMS have -> CAMS has

P9, L15: those for -> that of

P10, L14: tale -> tail (Please change later instances as well.)

P10, L25: West Sea -> the West Sea

P11, L28: than in Korea -> as in Korea

P12, L1: West Sea -> the West Sea

P13, L27: West Sea -> the West Sea

P14, L11: Baengnyeong -> the Baengnyeong

P15, L10 (and other locations): Olympic Park should always be capitalized (both words)

P15, L13: exhibit -> exhibits

P15, L23: shop tracks -> ship tracks

P17, L7-9: There are a few disjointed short sentences here. (e.g. "Because the size of CO data () is much larger than that of CO2 ().") Perhaps they could be joined together to make more sense?

P17, L24-25: near Korean coast -> near the Korean coast

P18, L15: "(by -2 to -4 ppmv for CO2 and -86 to -88 ppbv)" -> "(by -2 to -4 ppmv for CO2 and -86 to -88 ppbv for CO)"

P27, caption label: I would suggest using "bright" instead of "luminous" to describe the colours. Also add some articles when describing the sites, i.e. "The Olympic Park and Yonsei sites are located in an urban region (Seoul) while the Baengnyeong and Fukue sites are located in remote regions. The Taehwa site is located in a forest near Seoul."

P34, Figure 9: The figure label includes DC-8 still, but I believe this is in fact surface-based in situ data. If so, please remove these confusing labels.

P40, Table 3: The label refers to satellite measurements, but it should be in situ measurements.
* * *

---

## Author Comment (AC1) · 11 Jun 2018

The authors have presented an evaluation of the CAMS prediction system, focusing on CO and CO2, during the KORUS-AQ campaign. They evaluated three different CO and CO2 forecast and analysis products: 16-km CO and CO2 forecasts, 9-km CO and CO2 forecasts, and analyses of CO and CO2 at 80 km and 40 km, respectively. The CAMS products were compared to the KORUS-AQ aircraft data as well as to ground-based and satellite measurements of CO and CO2. They found that CAM overestimated CO2, suggesting a positive bias in background CO2, whereas it underestimated CO, with the underestimate confined mainly to the lower troposphere. The authors also found that CAMS underestimates the outflow of pollution from China, possibly due to an underestimate of Chinese emissions. The study is a nice evaluation of CAMS CO and CO2 under unique conditions. I have no major concerns about the analysis.

Response: Thank you!

My main concern is about the appropriateness of the manuscript for ACP. As a model evaluation study, I think it is better suited for GMD than ACP. My comments below are relatively minor, but must be address before the manuscript can be accepted for publication, if the Editor decides it is suitable for ACP.

Response: Thank you. We understand the reviewer's concern. But we think that our manuscript is still within the scope of ACP for the following reasons. We double-checked the main subject areas of ACP, which includes atmospheric modelling, field measurements, remote sensing, and laboratory studies of gases, aerosols, clouds and precipitation, radiation, and so on. In this study, we assess the performance of the Copernicus Atmosphere Monitoring Service (CAMS) global prediction system using field measurements from aircraft, ground sites, and ships, and remote sensing data during the KORUS-AQ field campaign. In addition to model evaluation, this study also addresses a few scientific topics on atmospheric chemistry and physics, including (1) anthropogenic combustion characteristics in Korea and China (as well as how well CAMS captures it), (2) impacts of different model configurations and environmental conditions on CO simulations, and (3) implications for CO emissions in CAMS. Thus, this manuscript is in line with the research focuses of ACP. Furthermore, we believe that the findings of this manuscript will be of interest not only to CAMS developers and users, but also to the general atmospheric chemistry community. Therefore, ACP is a perfect platform for us to share these results with the community. We sincerely hope that the editor will consider publishing this manuscript in ACP.

Comments
1. There is no mention of the **CAMS OH field**, which is critical for the simulation of CO. What is the global mean OH from the analyses and forecasts? On page 5, lines 28, it is mentioned that the 16-km CO forecasts use a linear chemistry scheme. A brief description of the scheme, either in the manuscript or in the supplement, would be helpful.

Response: The two high-resolution forecast products (FC16s and FC9s) employ a linear chemistry scheme, without the direct use of model OH.

The OH fields are only used in the CAMS ANs for CO which has full chemistry. In the ANs, the global and Northern Hemisphere mean are $0.98 \times 10^{-6}$ molecules/cm$^3$ and $1.20 \times 10^{-6}$ molecules/cm$^3$ during May 2016, respectively. We have extended description of the linear chemistry scheme at the end of Section 2.1. Specifically, we extended

"*A linear chemistry scheme is used in FC16s for CO (C-IFS-LINCO) for computationally expediency (Claeyman et al., 2010; Flemming et al., 2012; Massart et al. 2015; Eskes et al., 2017). Key aspects of the three CAMS configurations evaluated in this study are listed in Table 1.*"

to:

"*ANs for CO use the on-line implemented chemical mechanism (C-IFS-CB05, Flemming et al., 2015) that is an extended version of the Carbon Bond mechanism 5 (CB05, Yarwood et al., 2005). Because hydroxyl radical (OH) is an important sink for CO, modeled OH is critical for the simulation of CO (Gaubert et al., 2016, 2017). In the ANs for CO, the global and NH mean of air mass-weighted OH are $0.98 \times 10^{-6}$ molecules/cm$^3$ and $1.20 \times 10^{-6}$ molecules/cm$^3$ during May 2016, respectively (calculated following recommendations from Lawrence et al. (2001)). The mean OH from the ANs for CO is consistent with previous studies (e.g., Lawrence et al., 2001; Lelieveld et al., 2016; Gaubert et al., 2016, 2017). A linear chemistry scheme is (C-IFS-LINCO) used in FC16s and FC9s for CO for computationally expediency (Claeyman et al., 2010; Flemming et al., 2012; Massart et al. 2015; Eskes et al., 2017). C-IFS-LINCO computes CO sources and sinks using the approach developed by Cariolle and Déqué (1986) and updated by Cariolle and Teyssèdre (2007), without direct use of modeled OH. C-IFS-LINCO is less computationally demanding than the full chemistry, permitting simulations at higher resolutions (Massart et al. 2015). Key aspects of the three CAMS configurations evaluated in this study are listed in Table 1.*"

2. It is stated that the overestimate in CO2 is associated with the bias correction in the biogenic source of CO2, but there is no discussion of this "bias correction". Furthermore, in Agusti-Panareda et al. (2016) CAMS CO2 was underestimating CO2 observations from the surface in situ network and from TCCON, which the "bias correction" (the biospheric flux adjustment) reduced. Why is CAMS overestimating CO2 here? A discussion is needed about the **treatment of the biospheric fluxes in CAMS and its possible impact on the modeled CO2 over Korea**.

Response: First of all, we appreciate the reviewer for noticing this. According to Agusti-Panareda et al (2016), in the Northern Hemisphere there is a growing overestimation of the atmospheric CO$_2$ at the end of winter and throughout spring (from March to May); while at the end of the growing season in both the Northern Hemisphere and the Southern Hemisphere (August and March, respectively) there is a growing negative bias, i.e. an overestimation of the sink based on observations from NOAA/ESRL and TCCON (Section 5.1 of Agusti-Panareda et al (2016)). This is consistent with our finding. Agusti-Panareda et al (2016) also implies that the CO$_2$ overestimation by CAMS is enhanced in the BFAS simulation (Section 5.1 of Agusti-Panareda et al (2016)).

However, we note that the statement "*As found by Agusti-Panareda et al (2016), the overall overestimation of CO$_2$ is associated with the biogenic bias correction*" is inappropriate. We have changed it to "*Agusti-Panareda et al. [2016] also suggests CO$_2$ is overestimated by CAMS in the Northern Hemisphere at the end of winter and throughout spring*".

We have also included more discussions on the bias correction section 2.1 (where we introduce biogenic flux adjustment scheme (BFAS)), including treatment of the biospheric fluxes in CAMS, and its possible impact on the modeled CO$_2$ over Korea.

3. On page 11 it was shown that the model produced steeper vertical gradients in CO and CO2 than observed over Seoul, which the authors suggested may be due to weak boundary layer mixing. Since CAMS seems to perform better over Taehwa, it would be interesting to compare the vertical

gradients over Seoul and Taehwa in CAMS and in the observations to see if the issue is mainly an inability of CAMS to capture the PBL heights over the Seoul urban environment.

Response: We compared the observed vertical gradient of $CO_2$ over Seoul and Taehwa. We also analyzed the PBL height derived from observations and CAMS.

We have added the following analysis in the revised manuscript (Section 3.3.1):

"*We further find that compared with the Seoul metropolitan, the observed vertical gradient of $CO_2$ over Taehwa (~0.03 ppmv/hPa) below 925 hPa is smaller, which is relatively better captured by CAMS (0.02–0.12 ppmv/hPa). This again implies the possible inefficient boundary layer mixing in CAMS over the Seoul urban environment.*"

"*CO over Taehwa is more likely to be due to regional transport, as Taehwa is not a strong CO source region. Thus, the vertical gradient of CO over Taehwa does not necessarily reflect the impact of boundary layer mixing over Taehwa.*"

4. I am surprised that the analyses are not much better than the forecasts. Indeed, it seems as though the 9-km forecast is better than the analyses in some cases. I think it would be helpful for the reader if the authors expanded the description of the analyses to give the reader **more information about the configure and quality of the analyses.** Figure S1 and the brief text on page 5 are not enough.

Response: Analyses of both CO and $CO_2$ (ANs) do show improvement from the free running simulation (i.e., FC16s). However, the new 9-km forecast product (i.e., FC9s) is expected to have a better performance of FC9s because:

(1) FC9s is initialized with the analysis product every 24 hours (i.e., it incorporates information of analyses every 24 hours);

(2) FC9s has a much higher horizontal resolution (9 km) than the analyses (80 km for CO and 40 km for $CO_2$).

In addition to Figure S1 (i.e., the new Figure S2), Table 1 also summarized configurations of CAMS global atmospheric composition products including the analysis product of CO (AN_CHEM) and $CO_2$ (AN_GHG). We have also added more information about the configure and quality of the analyses:

"*Observations of both CO and $CO_2$ are assimilated in 12-hour assimilation windows. Inness et al. (2015) found that CO total column field, vertical distribution, and concentrations in the lower troposphere are improved by assimilating the CO total column from MOPITT. Assimilation of the GOSAT $XCO_2$ lead to improvements in mean absolute error and bias variability in $XCO_2$ fields during the year 2013 (Massart et al., 2016).*"

5. The discussion of **enhancement ratios** is confusing. It is unclear if the authors are using the slope of the CO/CO2 relationship or the slope of delta CO/delta CO2 relationship. The two approaches are different. The description in the text suggests that they are using the RMA regression of CO/CO2 to assess the combustion sources, but throughout the text there is use of the delta CO/delta CO2 notation. If they are indeed calculating an enhancement ratio (delta CO/delta CO2) above the background, how is the background being calculated? How sensitive is the analysis to the definition of the background?

Response: Thank you for pointing this out. We used slopes of the CO to $CO_2$ regressions as our enhancement ratios.

The estimated regression slope in the RMA corresponds to enhancement ratio of CO and $CO_2$ ($dCO/dCO_2$). No background values are used here. In fact, the definition of the background does not change the regression slope. Please see the following figure for a demonstration.

[Figure]

Nevertheless, we agree with the reviewer that the usage of $\Delta CO/\Delta CO_2$ in this manuscript could be misleading, and have changed $\Delta CO/\Delta CO_2$ to $dCO/dCO_2$.

6. The authors found that CAMS underestimated CO during China outflow events, but overestimated it under normal conditions. What are the **different source regions** for air reach the West Sea during "outflow" and "normal" conditions? To what degree is the model bias due to CAMS not capturing this difference in transport as compared to it not have the correct balance of emissions in China?

Our recent work with Community Atmosphere Model with chemistry (CAM-chem) tagged CO tracers studied the different source regions. Taking condition on June 5[th] (corresponding to the June 4[th] flight) as an example of normal conditions, both China and Korea contribute to the CO over the West Sea at surface. At 800 hPa, Japan, Russia, China, the rest of the world, and ocean all contribute to CO over the West Sea. However, CO concentrations over the West Sea is relatively low at these conditions. The following figure (Fig. S6 of Tang et al., 2018) shows spatial distributions of the tagged CO (ppbv) on June 5[th], 2016 at model surface, 800 hPa, and 500 hPa (Tag 1: Korea; Tag 2: Japan+Russia; Tag 3: Indonesia+India; Tag 4: EA-S; Tag 5: EA-M; Tag 6: EA-N; Tag 7: the rest of the world+ocean; Tag 8: $CH_4$ oxidation; Tag 9: biogenic; Tag 10: chemical production besides $CH_4$):

[Figure]

During outflow events (e.g., conditions during the May 30th flight), contribution from China are largely enhanced and becomes dominant at surface, 800 hPa, and 500 hPa. CO concentrations over the West Sea is relatively high during China outflow events. The following figure (Fig. S7 of Tang et al., 2018) shows spatial distributions of the tagged CO (ppbv) on May 31st, 2016 (corresponding to the May 30th flight) at model surface, 800 hPa, and 500 hPa (the Tags are the same as in the Fig. S6).

[Figure]

In Tang et al. (2018), we found that estimates of CO emissions in China rather than transport is potentially the main source of model bias over the West Sea. We have included the discussions and reference in the revised manuscript:

"*More elaborate analysis of source contributions during KORUS-AQ is beyond the scope of this study and can be found in Tang et al. (2018), which suggested that during China outflow events, the contribution from Chinese direct emissions to CO over the West Sea is largely enhanced and dominant.*"

Technical Comments
1. Page 4, line 26: add a comma between "forecast" and "CO2".
Response: We have added the comma.

2. Page 4, line 28: add "the" between "on" and "free-running".
Response: Thank you. We have edited accordingly.

3. Page 5, line 1: Is Figure 1 really necessary? I don't think it adds much to the manuscript. Since there are already 11 figures, I would suggest removing Figure 1.
Response: We have moved Figure 1 to supplement (the New Figure S2).

4. Page 6, lines 1 and 2; add "the" before "South Korean peninsula".
Response: Correction made.

5. Page 6, line 7: remove "including" before the list of the three questions.
Response: Thank you. We have deleted the "*including*".

6. Page 6, line 8: The wording for question (3) needs improving. The English is not quite correct.
Response: Thank you for pointing this out. We have changed the sentence to *"(3) how well do models perform and what improvements are needed to better represent atmospheric composition over Korea and its connection to the larger global atmosphere (Kim and Park, 2014, KORUS-AQ White Paper)."*

7. Page 8, lines 22 and 23: The revisit and overpass times seem to be used interchangeably here. The revisit time of OCO-2, for example, is 16 days since it is in the A-Train orbit. However, the local overpass time is around 1:30 pm. For GOSAT the revisit time is 3 days.
Response: Thank you. We have changed "*revisit*" to "*overpass*".

8. Page 8, line 26: Change 0.09e18 from e-notation to standard SI notation.
Response: We have changed the notation accordingly.

9. Page 9, line 5: Please add "is" between "CO2" and "associated".
Response: We have deleted this sentence.

10. Page 10, line 1: The variance in CO in the May 3rd data does not seem larger than average to me. In fact, it seems to be smaller than average.
Response: We have changed the sentences "*For example, the flights in May 3rd, May 17th, May 24th, May 29th, and May 30th were specifically designed to capture Chinese pollution outflow. In these days, the variances in CAMS biases for CO (but not $CO_2$) are larger than the average*"
to
"*For example, parts of flight tracks on May 3rd, May 17th, May 24th, May 29th, and May 30th were specifically designed to capture Chinese pollution outflow. In these days, the variances in CAMS biases for CO (but not $CO_2$) are generally larger than the average except for the flight tracks on May 3rd when Chinese influences were expected to be weak*"

11. Page 10, lines 12 and 14: Please change "tale" to "tail".
Response: Thank you. We have changed it.

12. Page 12, line 4: It is unclear what is meant by the statement that "the wind speeds dominate the transport flux variations in CO2." It the argument here that the meteorological uncertainty is the dominant contribution to the uncertainty in the forecast and analysis fields? If so, how does one come to that conclusion from Figure 7?
Response: Because the variations in $CO_2$ density are very low relative to $CO_2$ background, the pattern of the $CO_2$ fluxes in the Figure ($CO_2$ fluxes = wind speed × $CO_2$ density) mostly display pattern of wind speed instead of pattern of $CO_2$ density. However, we find the implication of "*the wind speeds dominate the transport flux variations in $CO_2$*" to be redundant with previous sentences, and this sentence itself is confusing. We have deleted it.

13. Page 15, line 23: Section 4.2 only discusses the comparison to ship data of CO, not CO and CO2.

Response: Thank you. We have deleted "*and CO₂*".

14. Page 15, line 23: change "shop tracks" to "ship tracks".
Response: Correction made.

15. Page 17, line 7: Change "size of CO data" to "amount of CO data".
Thank you. We have changed "*size*" to "*amount*".

16. Page 40, Table 3: The title for the table is wrong. This is the same title as for Table 4.
Response: Thank you for pointing this out. We have changed the title to "***Table 3.*** *Enhancement ratios of CO to CO₂ (ppbv/ppmv), CO and CO₂ correlations, and bias of CO to bias of CO₂ correlations from airborne measurements, CAMS FC16s, ANs, and FC9s.*"

17. Figure S1: Should the labels "FX9s" and "FX16s" in the figure be "FC9s" and "FC16s"?
Response: Thank you for noticing this. The labels should be "*FC9s*" and "*FC16s*", and we have corrected them.

---

## Author Comment (AC2) · 11 Jun 2018

This manuscript presents a comparison between three modelling products of the Copernicus Atmosphere Monitoring Service (CAMS) and in situ measurements from the KORUS-AQ (and KORUS-OC) campaign in the vicinity of the Korean peninsula during May/June 2016. Airborne, surface, ship-based, and satellite measurements of CO and CO2 are compared to the CAMS analysis and two forecast products at different spatial resolutions. The statistical analysis is relatively straightforward and clearly laid out, and some patterns of over- and underestimation are found for the two tracers under different conditions. The importance of vertical transport in the understanding of these differences could be further explored, as outlined below.

While I understand the other reviewer's comment that this manuscript might be a better fit for GMD, as it is assessing the capabilities of a specific modelling system, the general conclusions about the potential underestimation of CO emissions from China make it relevant for a broader audience as well. This is ultimately an editorial decision. However the quality of the manuscript, datasets and analysis is good, and appropriate for publication.
Response: Thank you!

Below are some suggestions, some major, some minor, on how the analysis might be slightly extended in order to better understand the processes driving the model-data mismatch.

Section 3.3: In the discussion about the **relative agreement in the profile for CO2** vs. **the disagreement between the lower atmosphere values between the observations and the model**, a discussion of the relevance of the mixing height and/or **planetary boundary layer** was somewhat lacking. A difference in profile shape can be attributed **to incorrect fluxes, incorrect mixing**, or a combination of the two. By having two tracers with differing results, it should be possible to deepen this analysis a bit.
There is further discussion about the vertical gradients of the tracers, but no attempt is made to diagnose the **PBL height**. Given the model data and the meteorological information from the aircraft profiles, this should be possible. Could you at least comment on this, and why such an approach was not undertaken? It is even suggested that there might be a "**possible weaker boundary layer mixing in CAMS**". Here **diagnosing the PBL height (as a function of time) from both the model fields and the profiles might be enlightening**.
Response: We have added the following discussion of analyzing differences in profile shapes by having two tracers in Section 3.3: "*CO over Taehwa is more likely to be due to regional transport, as Taehwa is not a strong CO source region. Thus, the vertical gradient of CO over Taehwa does not necessarily reflect the impact of BL mixing over Taehwa.*"
We have also added analyses and discussions of the relevance of the mixing height and/or planetary boundary layer, by adding the new Figure S7 (time series of the mixing layer heights from both the model fields and the measurements), and the following discussion:
"*We further compared the mixing layer (ML) height derived from the KORUS-AQ airborne DIAL-HSRL measurements of aerosol backscatter following the technique from Brooks et al. (2003), and the BL heights from CAMS. We note that ML height is only approximately equal to BL height. We find that CAMS generally underestimates BL heights during KORUS-AQ (Fig. S6). The model underestimation of BL over the Seoul metropolitan (-761.3±39.7 m) is stronger than that over Taehwa (721.7±38.6 m) which is covered by forests instead of urban. This is consistent with the*

*CAMS's relatively better capability of capturing vertical gradient of $CO_2$ over Taehwa compared to that over Seoul, supporting our previous implication of the possible inefficient BL mixing in CAMS over the Seoul urban environment.*"

Another interesting point might be the representation of urban effects for Seoul in particular. Here it would be interesting to compare the **PBL height** as modelled vs. measured in the vicinity of Seoul compared to other less rural sites. However, this may be beyond the scope of this study.
Response: Thank you. We compared modeled profiles of CO and $CO_2$ over Seoul and Taehwa to imply possible inefficient modeled boundary layer mixing over the Seoul metropolitan. Please see the response to the previous comment and the response to Reviewer #1, Comment 3 for details. We also added Figure S8, Model bias of boundary layer heights over Seoul and Taehwa (a less rural site).

For the special case of Seoul, the low altitude measurements were taken during missed approaches at the airport. Given all the air traffic in the region, might it be that the CO in this area is locally very much enhanced, and as such not representative of even the relatively small spatial footprint of the CAMS model? Here perhaps a referral to a relevant paper by Boschetti et al. (Tellus B, 2015) looking at **enhancements of CO in the boundary layer** from commercial airline measurements might be relevant.
Response: Thank you. We have added the reference (Boschetti et al., 2015) as well as the following discussion in the Section 3.3.1:
"*In addition, given the air traffic over the Seoul Air Base (where the DC-8 aircraft frequently conducted missed approaches), emissions from airplanes may also contribute to the model biases.*"

Regarding the assessment of the outflow over the West Sea, I was confused by the phrase: "Hence, the wind speeds dominate the transport flux variations in CO2." I'm not sure what is meant here. Is this because the outflow pattern wasn't as strong as for CO? But aren't both flux variations (more or less) linearly dependent on wind speed anyhow? Please clarify.
Response: Thank you. We have deleted this sentence. Please see the response to Reviewer #1, Comment 12 for details.

The discussion about the correlation of CO and CO2 over the West Sea is quite interesting, and invites further inquiry. The suggestion that the difference in time factors for anthropogenic CO and CO2 (with the former having constant monthly values and the latter having diurnal variability) should effect the correlation over Korea as well. Could it be explained by the differences in transport times, e.g. diurnal CO2 emissions peak in daytime while measurements are being made over Korea, whereas daytime measurements over the West Sea represent nighttime emissions from China, where the difference in time factors is at a maximum? In terms of just the correlation in the fluxes, it should be easy to test if EDGAR has a higher spatial correlation between CO2 and CO in Korea vs. China.
Response: Thank you. The high observed CO and $CO_2$ correlations over the West Sea and Seoul suggest that CO and $CO_2$ are likely from common sources. We found that the difference in the time factors of CO emissions and $CO_2$ fluxes may contribute to the model's inability of capturing such high correlation over the West Sea. However, such difference in the time factors is unlikely to impact the modeled correlation over Seoul as much as it impacts the West Sea. In fact, the diurnal variability of $CO_2$ fluxes comes from the $CO_2$ net ecosystem exchange rather than the

anthropogenic part. Since the flights over Seoul are close to the strong common anthropogenic sources of CO and $CO_2$ (i.e., the Seoul metropolitan area), the correlation over Seoul is dominated by anthropogenic emissions and unlikely impacted by diurnal variability of $CO_2$ fluxes that comes from the $CO_2$ net ecosystem exchange. This is supported by the consistency between observed and modeled correlations.

As for the other 3 groups over Korea (i.e., Taehwa, Seoul-Jeju jetway, Seoul-Busan jetway), their observed correlations are not high (i.e., 0.68, 0.62, 0.60, respectively) at the first place compared to the observed high correlations over the West Sea (0.89) and Seoul (0.78). This implies that $CO_2$ and CO observed over these three flight groups may not come from common sources and/or have been mixed with the environment.

We agree with the reviewer that correlation in the fluxes may provide valuable insights to explain the correlations in the modeled abundance correlations. The following figure shows that time series of spatial correlations between CO emissions and $CO_2$ fluxes in CAMS over East China (which dominates Chinese contribution to the West Sea (Tang et al., 2018)) and Korea. There is a strong diurnal cycle in the correlations caused by the difference in time factors.

[Figure]

The diurnal cycle of spatial correlations between CO emissions and $CO_2$ fluxes over Korea in CAMS peaks (~0.7) in daytime while measurements over Korea were made. On the other hand, during the nighttime, the correlations between CO emissions and $CO_2$ fluxes in CAMS are relatively low over East China (<0.4). This implies that the relatively low correlations between the CO and $CO_2$ abundances over the West Sea in CAMS may reflect the effect of nighttime emissions from East China in CAMS.

We thank the reviewer for pointing this out and have included this in the manuscript (text in Section 3.4 and the new Fig. S8).

The analysis of the satellite data is not particularly illuminating, with the exception of the separation of MOPITT data into those influence by outflow. Regarding the use of the OCO-2 data, most of the data references are pre-launch, and should be updated. Wunch et al. 2017 would be a better up to date reference than those from 2011, and an updated estimate of the OCO-2 precision, even if it is coming from grey literature (such as the ACOS OCO-2 User's Guide) would be preferable to a largely theoretical assessment from Boesch et al., 2011.

Response: Thank you. We have included the following two references:

*Osterman, G. B., Eldering, A., Avis, C., Chafin, B., O'Dell, C., Frankenberg, C., ... & Crisp, D. (2015). Orbiting Carbon Observatory-2 (OCO-2) Data Product User's Guide, Operational L1 and L2 Data Versions 7 and 7R. NASA Jet Propulsion Laboratory, California Institute of Technology.*
*Wunch, D., Wennberg, P. O., Osterman, G., Fisher, B., Naylor, B., Roehl, C. M., ... & Griffith, D. W. (2017). Comparisons of the Orbiting Carbon Observatory-2 (OCO-2) X CO 2 measurements with TCCON. Atmospheric Measurement Techniques, 10(6), 2209.*

It is unclear what is meant by the "recommended quality control" in section 2.2.4. Does this mean the standard quality flag? Was the bias correction applied? Was a certain warn-level threshold used? Please elaborate.

Response: "recommended quality control" means the standard quality flag, and we have changed the term in the manuscript. The standard quality flag we used is from Table 1 and Table 2 of Mandrake et al. (2015) (https://co2.jpl.nasa.gov/static/docs/OCO2_XCO2_Lite_Files_and_Bias_Correction_0716.docx):

**Table 1: Quality Filters Applied to Land Soundings**

| All Land Soundings | | |
|---|---|---|
| Field | Lower Limit ( > or =) | Upper Limit ( < or = ) |
| Warn level | N/A | 15 |
| Outcome flag (not in lite file) | N/A | 2 |
| Preprocessors/h2o_ratio | 0.700 | 1.030 |
| Preprocessors/co2_ratio | 0.995 | 1.025 |
| Preprocessors/dp_apb | -15.00 | 5.00 |
| Retrieval/dp | -5.00 | 10.0 |
| Retrieval/aod_ice | N/A | 0.050 |
| Retrieval/Aod_sulfate | N/A | 0.400 |
| Retrieval/Aod_dust* | 0.001 | 0.30 |
| Retrieval/Co2_grad_del | -70.0 | 70.0 |
| Retrieval/albedo_2 | 0.10 | N/A |
| Blended albedo (2.4*albedo_3 – 1.13*albedo_1) (both in retrieval group) | N/A | 0.8 |
| dof_co2 (not in lite product) | 1.8 | N/A |
| Sounding/airmass | N/A | 3.6 |
| * or AOD dust = 0.0 | | |

**Table 2: Quality Filters Applied to Ocean Glint Soundings**

| Ocean Glint Soundings | | |
|---|---|---|
| Field | Lower Limit ( > or =) | Upper Limit ( < or = ) |
| Warn level | N/A | 15 |
| Outcome flag (not in lite file) | N/A | 2 |
| Preprocessors/co2_ratio | 0.994 | 1.020 |
| Preprocessors/dp_apb | N/A | 0.00 |

| Retrieval/dp | -3.00 | 9.0 |
|---|---|---|
| Retrieval/Co2_grad_del | -30.0 | 5.0 |
| Retrieval/albedo_slope_3•10$^5$ | 1.0 | 10.0 |
| Retrieval/windspeed | 2.0 | N/a |
| Sounding/snr_weak_co2 | 380 | N/A |
| Sounding/airmass | N/A | 3.5 |

We used the suggested warn-level threshold in the Table (i.e., <=15). We have added the warn_level information in the manuscript. The bias correction is not applied to the data used in this study, as we used Standard Data files (L2Std) instead of Lite files.

If the Taylor skill score is being used for the assessment of the forecasting skill as in section 3.1, the equation should be in the main paper, and not just in the suppplement. Please include it here as well.
Response: We have moved the equation from supplement to the section 3.1 of the paper.

P4, L18-20: The text here states that the CO analysis runs at "approximately 40 km horizontal resolution", but in Figure 1 it is shown to be 80 km horizontal resolution. Later on page 5 80 km is given again, and the text on P4 refers to that fact that the CO2 analysis is at a higher spatial resolution (in both the horizontal and vertical). Please ensure that the information is consistent and correct.
Response: Thank you for pointing it out. The CO analysis is at approximately 80 km horizontal resolution while the $CO_2$ analysis is at the approximately 40 km horizontal resolution. We have corrected the P4, L18-20.

Minor/typographical comments:
P2, L11: show -> shows
Response: We have corrected it.

P2, L12: "over Seoul metropolitan" -> either "over the Seoul metropolitan area" or "over Seoul"
Response: Thank you. We have changed that to "*over the Seoul metropolitan area*".

P3, L6: near-real time -> near-real-time
Response: Correction made.

P3, L16: field -> field campaign
Response: Thank you. We have added "*campaign*" there.

P4, L20: Perhaps this should be one sentence?
Response: Thank you for noticing this. This should be one sentence and we have corrected it.

P4, L25: 4-days shouldn't be hyphenated (four days)
Response: Correction made.

P4, L26: 16km -> 16 km

Response: Thank you for pointing this out. We noticed that in the text we used "*16km forecast*" or "*16 km forecast*" a few times. To be consistent, we changed them into "*16-km forecast*". Same for the "*9-km forecast*".

P5, L26: The -> the

Response: Correction made.

P6, L1, L2, and often afterwards: South Korean peninsula -> the South Korean peninsula

Response: Thank you. We have edited the manuscript accordingly.

P6, L8-10: The third scientific question needs to be restated. It doesn't make sense as it is written here.

Response: Thank you. We have rephrased the question (3):

*"(3) how well do models perform and what improvements are needed to better represent atmospheric composition over Korea and its connection to the larger global atmosphere (Kim and Park, 2014, KORUS-AQ White Paper)."*

P6, L27: data is -> data are

Response: Correction made.

P6, L30-31: Wouldn't UTC be one day behind local time?

Response: Thank you for pointing this out. Korea time = UTC time +9. We have changed "*UTC time is one day ahead of Korea local time*" to "*UTC time is one day behind Korea local time*".

P7, L7: combustion signatureS (plural needed to match grammar)

Response: Correction made.

P7, L10-12: Not sure which preposition should be used to describe the jetway flights, I would suggest "in", but consistency is more important. Also check the grammar: "Flights in the Seoul-Busan jetway were designed to capture... Flight in the Seoul-Jeju jetway, on the other hand, sampled air over..."

Response: We have changed "*Flights over Seoul-Busan jetway*" to "*Flights in Seoul-Busan jetway*"; and "*The flights in Seoul-Jeju jetway, on the other hand, samples air over local power*" to "*The flights in Seoul-Jeju jetway, on the other hand, sample air over local power*".

P7, L17: Baengnyeong site is located in less populated Baengnyeong Island, Incheon which is northwest of Seoul. -> The Baengnyeong site is located on the sparsely populated Baengnyeong Island, Incheon, northwest of Seoul.

Response: Thank you. We have changed accordingly.

P7, L19: "on remote" -> "on the remote"

Response: Correction made.

P8, L21: resolutions -> the resolutions

Response: Correction made.

P9, L2: Here is the first of many instances of referring to the in situ measurements collected from the DC-8 aircraft as simply "DC-8". As a reader I found this jarring. Perhaps instead refer to the dataset as the "DC-8 in situ data" or the "aircraft data" or "the airborne measurements"?
Response: Thank you for pointing it out. We have changed "*DC-8*" to "*the DC-8 aircraft data*" or "*the airborne measurements*" in the manuscript to refer the airborne measurements from the DC-8 aircraft.

P9, L8: inconsistent description of correlation range (to vs. -)
Response: We have changed the "*to*" to "*–*".

P9, L12: CAMS have -> CAMS has
Response: Correction made.

P9, L15: those for -> that of
Response: Correction made.

P10, L14: tale -> tail (Please change later instances as well.)
Response: Correction made.

P10, L25: West Sea -> the West Sea
Response: Thank you. We have changed all the "*West Sea*" to "*the West Sea*" in the revised manuscript.

P11, L28: than in Korea -> as in Korea
Response: Correction made.
P12, L1: West Sea -> the West Sea
Response: Correction made.

P13, L27: West Sea -> the West Sea
Response: Correction made.

P14, L11: Baengnyeong -> the Baengnyeong
Response: Correction made.

P15, L10 (and other locations): Olympic Park should always be capitalized (both words)
Response: Thank you. Correction made.

P15, L13: exhibit -> exhibits
Response: Correction made.

P15, L23: shop tracks -> ship tracks
Response: Correction made.

P17, L7-9: There are a few disjointed short sentences here. (e.g. "Because the size of CO data () is much larger than that of $CO_2$ ().") Perhaps they could be joined together to make more sense?
Response: We have joined this sentence and the two followed sentences into one. I.e., we have

changed:

*"Because the amount of CO data (13612 retrievals for MOPITT and 25509 for IASI over our study domain during KORUS-AQ) is much larger than that of $CO_2$ (42 for GOSAT over our domain during KORUS-AQ). This is illustrated in Fig. 10 and listed in Table 4. There are more observational constraints for CO in CAMS resulting to better performance of ANs CO."*

to:

*"Because the amount of CO data (13612 retrievals for MOPITT and 25509 for IASI over our study domain during KORUS-AQ) is much larger than that of $CO_2$ (42 for GOSAT over our domain during KORUS-AQ), there are more observational constraints for CO in CAMS resulting to better performance of ANs CO (Fig. 10 and Table 4)."*

P17, L24-25: near Korean coast -> near the Korean coast
Response: Correction made.

P18, L15: "(by -2 to -4 ppmv for CO2 and -86 to -88 ppbv)" -> "(by -2 to -4 ppmv for CO2 and -86 to -88 ppbv for CO)"
Response: Correction made.

P27, caption label: I would suggest using "bright" instead of "luminous" to describe the colours. Also add some articles when describing the sites, i.e. "The Olympic Park and Yonsei sites are located in an urban region (Seoul) while the Baengnyeong and Fukue sites are located in remote regions. The Taehwa site is located in a forest near Seoul."
Response: Thank you. We have substituted *"luminous"* with *"bright"*. We have also added references to the caption when describing the sites.

P34, Figure 9: The figure label includes DC-8 still, but I believe this is in fact surfacebased in situ data. If so, please remove these confusing labels.
Response: Thank you for pointing this out. We have changed *"DC-8"* to *"Observations"* in the label.

P40, Table 3: The label refers to satellite measurements, but it should be in situ measurements.
Response: Thank you. Correction made.